# Lie Neurons: Adjoint-Equivariant Neural Networks for Semisimple Lie Algebras

## Abstract

This paper proposes an adjoint-equivariant neural network that takes Lie algebra data as input. Various types of equivariant neural networks have been proposed in the literature, which treat the input data as elements in a vector space carrying certain types of transformations. In comparison, we aim to process inputs that are transformations between vector spaces. The change of basis on transformation is described by conjugations, inducing the adjoint-equivariance relationship that our model is designed to capture. Leveraging the invariance property of the Killing form, the proposed network is a general framework that works for arbitrary semisimple Lie algebras. Our network possesses a simple structure that can be viewed as a Lie algebraic generalization of a multi-layer perceptron (MLP). This work extends the application of equivariant feature learning. As an example, we showcase its value in homography modeling using $\mathfrak{sl}(3)$ Lie algebra.

## 1 Introduction

In geometric problems such as control theory, robotics, computer vision and graphics, Lie group methods provide the machinery to study continuous symmetries inherent to the problem (Murray et al., 1994; Lynch & Park, 2017; Barrau & Bonnabel, 2017; van Goor et al., 2020; Liu et al., 2010; Yang et al., 2021). Lie algebras are vector spaces that locally preserve the group structure, enabling efficient computation (Teng et al., 2022). The standard group representation (linear group action) on Lie algebras is given by conjugation or adjoint action (Hall, 2013).

The equivariance property preserves the symmetry group structure, often a Lie group, such that the feature map commutes with the group representation. Equivariant models have gained success in various domains, including but not limited to the modeling of molecules (Thomas et al., 2018), physical systems (Finzi et al., 2020), social networks (Maron et al., 2018), images (Worrall et al., 2017), and point clouds (Zhu et al., 2023). Convolutional neural networks (CNNs) are translation-equivariant, enabling stable image features regardless of the pixel positions in the image plane. Typical extensions include rotation (Cohen et al., 2017) and scale (Worrall & Welling, 2019) equivariance, while more general extensions are also explored (MacDonald et al., 2022).

In this paper, we propose a new type of equivariant model that captures the symmetry in conjugacy classes. *We view the input data of our model not as elements in a vector space but as transformations (maps) between vector spaces, typically represented as matrices for linear groups.* Accordingly, group action on the input data stands for a change of basis on the transformations, which is also known as conjugation. The input data as transformations, when viewed as a continuous symmetry group, form a Lie group. To facilitate the numerical computation, the proposed network takes the transformations to the Lie algebra. The Lie algebraic approach is particularly attractive as it enables working in vector spaces and exploiting its isometry structure to handle data in typical vector form. One way to intuitively understand the relations between existing equivariant networks and our proposed network is illustrated in Figure 1.

Overall, the contributions of our paper are summarized as follows:

- We propose a new adjoint-equivariant network architecture, enabling the processing of input data that represent transformations.
- We develop new network designs using the Killing form and the Lie bracket structure for equivariant activation and invariant layers for Lie algebraic representation learning.

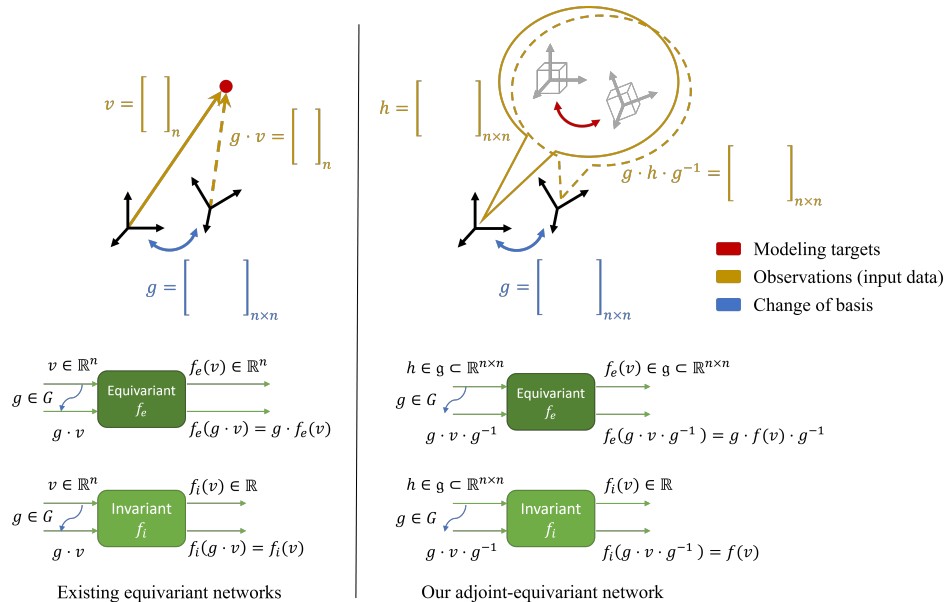

Figure 1: Comparison between existing equivariant networks and our work. Red represents the underlying objects to be studied by the models. For example, a commonly studies type of object in existing equivariant networks is shapes. For our work, the studied object is transformations. Given a reference frame, we obtain an observation of the studied object illustrated in yellow. For existing equivariant networks, the inputs are represented as vectors (including tensors). For our work, the inputs are represented as matrices. Change of basis, illustrated in blue, acts on vectors by left multiplication while acting on transformations by conjugation.

- The proposed network models the equivariance of any semisimple Lie algebras, which relaxes the requirement in previous work Deng et al. (2021) for the Lie group to be compact.
- The implementation will be open-sourced.

## 2 RELATED WORK

Equivariant networks enable the model output to change in a predicted way as the input goes through certain transformations. This means that the model, by construction, generalizes over the variations caused by those transformations. Therefore, it reduces the sampling complexity in learning and improves the robustness and transparency facing input variations. Cohen & Welling (2016a) initials the idea to generalize equivariance in deep learning models, realizing equivariance to 90-degree rotations in a 2D image plane using group convolution. This method works with discrete transformations by augmenting the input domain with a dimension for the set of transformations. The approach is generalized to other discretized groups in $SE(2)$, $SE(3)$, and $E(3)$ (Hoogeboom et al., 2018; Winkels & Cohen, 2018; Worrall & Brostow, 2018; Chen et al., 2021). Steerable convolution is proposed in Cohen & Welling (2016b), leveraging the irreducible representations to remove the need for discretization and facilitate equivariant convolution on continuous groups in the frequency domain (Worrall et al., 2017; Cohen et al., 2017; Weiler et al., 2018). Beyond convolutions, more general equivariant network architectures are proposed, for example, Fuchs et al. (2020); Hutchinson et al. (2021); Chatzipantazis et al. (2022) for transformers and Batzner et al. (2022); Brandstetter et al. (2021) for message passing networks. Vector Neurons (Deng et al., 2021) designs a multi-layer perception (MLP) and graph network that generalize the scalar features to 3D features to realize $SO(3)$-equivariance on spatial data. The above works mainly focus on compact groups, on which more general recipes for building equivariant layers that are not limited to a specific group are also proposed (Kondor & Trivedi, 2018; Cohen et al., 2019; Weiler & Cesa, 2019; Xu et al., 2022; Lang & Weiler, 2020). The extension of equivariance beyond compact groups is also explored. Finzi et al. (2021) constructs MLPs equivariant to arbitrary matrix groups using their finite-dimensional representations. With the Monte Carlo estimator, equivariant convolutions are generalized to matrix

groups with surjective exponential maps (Finzi et al., 2020) and all finite-dimensional Lie groups (MacDonald et al., 2022), where Lie algebra is used to parameterize elements in the continuous Lie groups as a lifted domain from the input space.

Our model structure resembles the MLP style of Vector Neurons (Deng et al., 2021), but our work models the equivariance of arbitrary semisimple groups under adjoint actions. Lie algebra is the input space of our network, representing transformation data.

## 3 PRELIMINARIES

We provide some preliminaries for Lie groups by focusing on matrix Lie groups. For detailed explanations, we refer the readers to Hall (2013); Rossmann (2006); Kirillov (2008).

### 3.1 LIE GROUP AND LIE ALGEBRA

A Lie group $\mathcal{G}$ is a smooth manifold whose elements satisfy the group axioms. Because of this, a special vector space naturally arises at the identity of every Lie group named the Lie algebra, denoted $\mathfrak{g}$. A Lie algebra locally captures the structure of the Lie group.

Every Lie algebra is equipped with an asymmetric binary operator called the Lie bracket:

$$[\cdot, \cdot] : \quad \mathfrak{g} \times \mathfrak{g} \to \mathfrak{g}. \tag{1}$$

The elements of the Lie algebra have non-trivial structures. However, since the Lie algebra is a vector space, for a Lie algebra of dimension $m$, we can always represent it using $\mathbb{R}^m$ given appropriate bases. As a result, we introduce two useful maps Chirikjian (2011):

$$\text{Vee} : \mathfrak{g} \to \mathbb{R}^m, \quad x^\wedge \mapsto (x^\wedge)^\vee = \sum_{i=1}^m x_i e_i, \quad \text{Hat} : \mathbb{R}^m \to \mathfrak{g}, \quad x \mapsto x^\wedge = \sum_{i=1}^m x_i E_i, \tag{2}$$

where $e_i$ are the canonical basis of $\mathbb{R}^m$ and $E_i = (e_i)^\wedge \in \mathfrak{g}$.

For example, the $\mathfrak{so}(3)$ elements are skew-symmetric, $\begin{bmatrix} 0 & -\omega_z & \omega_y \\ \omega_z & 0 & -\omega_x \\ -\omega_y & \omega_x & 0 \end{bmatrix} \in \mathfrak{so}(3)$. One can find the canonical basis as follows.

$$E_x = \begin{bmatrix} 0 & 0 & 0 \\ 0 & 0 & -1 \\ 0 & 1 & 0 \end{bmatrix}, E_y = \begin{bmatrix} 0 & 0 & 1 \\ 0 & 0 & 0 \\ -1 & 0 & 0 \end{bmatrix}, E_z = \begin{bmatrix} 0 & -1 & 0 \\ 1 & 0 & 0 \\ 0 & 0 & 0 \end{bmatrix}. \tag{3}$$

With $v = [\omega_x, \omega_y, \omega_z]^\mathsf{T} \in \mathbb{R}^3$, we have

$$(v)^\wedge = \omega_x E_x + \omega_y E_y + \omega_z E_z = \begin{bmatrix} 0 & -\omega_z & \omega_y \\ \omega_z & 0 & -\omega_x \\ -\omega_y & \omega_x & 0 \end{bmatrix} = W \in \mathfrak{so}(3), \tag{4}$$

$$(W)^\vee = v = [\omega_x, \omega_y, \omega_z]^\mathsf{T}. \tag{5}$$

Using the Hat and Vee maps, we can represent an element of the Lie algebra in a neural network using $\mathbb{R}^m$, while performing structure-preserving operations on $\mathfrak{g}$. In this work, we use the basis of $\mathfrak{sl}(3)$ from Winternitz (2004) to construct the Hat and Vee maps.

### 3.2 ADJOINT REPRESENTATION

Given an element of the Lie algebra $X \in \mathfrak{g}$ and its corresponding Lie group $\mathcal{G}$, every $a \in \mathcal{G}$ defines an automorphism of the Lie algebra $Ad_a : \mathfrak{g} \to \mathfrak{g}$ by $Ad_a(X) = aXa^{-1}$. This is called the adjoint action of the group $\mathcal{G}$ on the Lie algebra $\mathfrak{g}$. It represents the change of basis operations on the algebra. Since the adjoint $Ad_a$ is linear, we can find a matrix that maps the $\mathbb{R}^m$ representation of the Lie algebra to another. That is, for every $Ad_a$ and $X \in \mathfrak{g}$, we have

$$Adm_a : \quad \mathbb{R}^m \to \mathbb{R}^m, \quad x \mapsto Adm_a x, \tag{6}$$

with $Adm_a \in \mathbb{R}^{m \times m}$, $x^\wedge = X$ and $Adm_a x = (ax^\wedge a^{-1})^\vee$. This is an important property as it allows us to model the group adjoint action using a matrix multiplication on $\mathbb{R}^m$, which enables the adjoint equivariant layer design in Section 4.

Conversely, if we view the $Ad$ as a function of a group element $a \in \mathcal{G}$, it maps the group element to a Lie algebra automorphism:

$$Ad : \mathcal{G} \to \texttt{Aut}(\mathfrak{g}), \quad a \mapsto Ad_a. \tag{7}$$

This $Ad$ is called the *adjoint representation* of the group. Similarly, we can obtain the adjoint representation of the Lie algebra by differentiating the adjoint representation of the group at the identity:

$$ad : \mathfrak{g} \to \texttt{Der}(\mathfrak{g}), \quad X \mapsto ad_X(\cdot) = [X, \cdot], \tag{8}$$

where $\texttt{Der}(\mathfrak{g})$ is the Lie algebra of $\texttt{Aut}(\mathfrak{g})$, and $[\cdot, \cdot]$ is the Lie bracket of the Lie algebra. For a matrix group, the Lie bracket is defined by the commutator: $[X, Y] = XY - YX$. It is worth noticing that the Lie bracket is equivariant under the group adjoint action.

### 3.3 KILLING FORM

If a Lie algebra $\mathfrak{g}$ is of finite dimension and associated with a field $\mathbb{R}$, a symmetric bilinear form called the *Killing form* is defined as:

$$B(X, Y) : \mathfrak{g} \times \mathfrak{g} \to \mathbb{R}, \quad (X, Y) \mapsto \text{tr}(ad_X \circ ad_Y) \tag{9}$$

**Definition 1** *A bilinear form $B(X, Y)$ is said to be non-degenerate iff $B(X, Y) = 0$ for all $Y \in \mathfrak{g}$ implies $X = 0$.*

**Theorem 1 (Kirillov (2008))** *A Lie algebra is semisimple iff the Killing form is non-degenerate.*[1]

**Theorem 2 (Kirillov (2008))** *The Killing form is invariant under the group adjoint action $Ad_a$ for all $a \in \mathcal{G}$, i.e.,*

$$B(Ad_a \circ X, Ad_a \circ Y) = B(X, Y).$$

If the Lie group is also compact, the Killing form is negative definite, and the inner product naturally arises from the negative of the Killing form.

## 4 METHODOLOGY

We present Lie Neurons (LN), a general adjoint-equivariant neural network on Lie algebras. It is greatly inspired by Vector Neurons (VN) (Deng et al., 2021). Vector Neurons take 3-dimensional vectors as inputs, typically viewed as points in Euclidean space. The 3D Euclidean dimension is preserved in the features, independent from the feature channel dimension. In other words, Vector Neurons lift conventional $\mathbb{R}^C$ features to $\mathbb{R}^{3 \times C}$, allowing the same SO(3) actions to be applied in the input space and the feature space, facilitating the equivariance property.

The Lie Neurons generalize the 3-dimensional SO(3) equivariant VN to a $K$-dimensional network that is equivariant by construction for any semisimple Lie algebra. Different from the VN, which take points in Euclidean space as input, the Lie Neurons take elements of a Lie algebra as inputs, and they capture the equivariance of a Lie group under the adjoint action. We will discuss how the Lie Neuron almost specializes to the VN later in Section 4.5. Here, we start by describing the network structure of Lie Neurons.

The standard multilayer perceptron (MLP) networks are constructed with scalar neurons, $z \in \mathbb{R}$. For each layer, the neurons are concatenated in the feature dimension into a $C^{(d)}$-dim vector $\mathbf{z} \in \mathbb{R}^{C^{(d)}}$, where $d$ denotes the layer index. Vector Neurons lift the neuron representation from scalars to $\mathbb{R}^3$. For an input composed of a set of $N$ points (e.g., a point cloud), the features learned from a VN layer are of shape $\mathbb{R}^{3 \times C^{(d)} \times N}$.

The Lie Neurons generalize Vector Neurons. Each Lie Neuron, $X \in \mathfrak{g}$, is an element of a semisimple Lie algebra. A Lie algebra is a vector space with non-trivial structures. However, using (2), we can

---

[1]This is also known as the *Cartan's Criterion*.

express the neuron in $x = X^\vee \in \mathbb{R}^K$ with appropriate bases. Similar to Vector Neurons, the features learned from a Lie Neuron layer are $\mathcal{X}^{(d)} = \{\mathbf{x}_i\}_{i=1}^N \in \mathbb{R}^{K \times C^{(d)} \times N}$, where $\mathbf{x}_i \in \mathbb{R}^{K \times C^{(d)}}$.

By construction, the LN are equivariant to the group adjoint action (also known as similarity transform in matrix groups). In particular, for a given element $a$ in a semisimple Lie group $\mathcal{G}$, we have

$$f(a\mathcal{X}a^{-1}; \theta) = af(\mathcal{X}; \theta)a^{-1}, \tag{10}$$

where $f$ is the function defined by an LN model with parameters $\theta$. [2] The Lie Neurons framework consists of a linear layer, two nonlinear activation layers, a pooling layer, and an invariant layer. We start by discussing the linear layers as follows.

### 4.1 LINEAR LAYERS

Linear layers are the basic building blocks of an MLP. A linear layer has a learnable weight matrix $\mathbf{W} \in \mathbb{R}^{C \times C'}$, which operates on input features $\mathbf{x} \in \mathbb{R}^{K \times C}$ by right matrix multiplication:

$$\mathbf{x}' = f_{\text{LN-Lin}}(\mathbf{x}; \mathbf{W}) = \mathbf{x}\mathbf{W} \in \mathbb{R}^{K \times C'}. \tag{11}$$

Recall (6), if we use the vector representation $x \in \mathbb{R}^K$ for $X \in \mathfrak{g}$, we can always find a linear adjoint matrix $Adm_a \in \mathbb{R}^{K \times K}$ such that $Adm_a x = aXa^{-1}$. A linear layer can be viewed as a matrix multiplication on the right ($C$ dimension), while the adjoint matrix acts on the left ($K$ dimension). As a result, the adjoint action on the linear layer becomes:

$$
\begin{aligned}
f_{\text{LN-Lin}}(Ad_a(\mathbf{x}); \mathbf{W}) &= f_{\text{LN-Lin}}(Adm_a\mathbf{x}; \mathbf{W}) \\
&= Adm_a\mathbf{x}\mathbf{W} \in \mathbb{R}^{K \times C'} \\
&= Adm_a f_{\text{LN-Lin}}(\mathbf{x}; \mathbf{W}) \\
&= Ad_a(f_{\text{LN-Lin}}(\mathbf{x}; \mathbf{W})),
\end{aligned}
\tag{12}
$$

which proves the equivariant property of the linear layer. It is worth mentioning that we ignore the bias term to preserve the equivariance. Lastly, similar to the Vector Neurons, the weights may or may not be shared across the elements $\mathbf{x}$ of $\mathcal{X}$.

### 4.2 NONLINEAR LAYERS

Nonlinear layers enable the neural network to approximate complicated functions. We propose two designs for the equivariant nonlinear layers, `LN-ReLU` and `LN-Bracket`.

#### 4.2.1 LN-RELU: NONLINEARITY BASED ON THE KILLING FORM

We can use an invariant function to construct an equivariant nonlinear layer. The VN leverages the inner product in a standard vector space, which is invariant to $\text{SO}(3)$, to design a vector ReLU nonlinear layer. We generalize this idea by replacing the inner product with the negative of the Killing form. As described in Section 3, the negative Killing form falls back to inner product for compact semisimple Lie groups, and it is invariant to the group adjoint action.

For an input $\mathbf{x} \in \mathbb{R}^{K \times C}$, a Killing form $B(\cdot, \cdot)$, and a learnable weight $U \in \mathbb{R}^{C \times C}$, the nonlinear layer $f_{\text{LN-ReLU}}$ is defined as:

$$f_{\text{LN-ReLU}}(\mathbf{x}) = \begin{cases} \mathbf{x}, & \text{if } B(x, d) \leq 0 \\ \mathbf{x} + B(x, d)d, & \text{otherwise}, \end{cases} \tag{13}$$

where $d = \mathbf{x}U \in \mathbb{R}^{K \times C}$ is the learnable direction. Optionally, we can share the learned direction across channels by setting $U \in \mathbb{R}^{C \times 1}$.

From Theorem 2, we know the Killing form is invariant under the group adjoint action, and the equivariance of the learned direction is proven in (12). Therefore, the second output of (13) becomes a linear combination of two equivariant quantities. As a result, the nonlinear layer is equivariant to the adjoint action.

We can also construct variants of ReLU, such as the leaky ReLU in the following form:

$$f_{\text{LN-LeakyReLU}} = \alpha\mathbf{x} + (1 - \alpha)f_{\text{ReLU}}(\mathbf{x}). \tag{14}$$

---

[2] We slightly abuse the notation here by setting $a\mathcal{X}a^{-1} = \{\{(a(x_{ij})^\wedge a^{-1})^\vee\}_{i=1}^{C^{(d)}}\}_{j=1}^N$.

### 4.2.2 LN-BRACKET: NONLINEARITY BASED ON THE LIE BRACKET

As introduced in Section 3, Lie algebra is a vector space with an extra binary operator called Lie bracket, which is equivariant under group adjoint actions. For a matrix Lie group, the Lie bracket of its Lie algebra is defined using the commutator: $[X, Y] = XY - YX$. We use this operation to build a novel nonlinear layer.

We use two learnable weight matrices $U, V \in \mathbb{R}^{C \times C}$ to map the input to different Lie algebra vectors, $\mathbf{u} = \mathbf{x}U, \mathbf{v} = \mathbf{x}V$. The Lie bracket of $\mathbf{u}$ and $\mathbf{v}$ becomes a nonlinear function on the input: $\mathbf{x} \mapsto [(\mathbf{x}U)^\wedge, (\mathbf{x}V)^\wedge]$. Theoretically, we can directly use it as our nonlinear layer. However, we note that the Lie bracket essentially captures *the failure of matrices to commute*. (Guggenheimer, 2012), and that $[X, X] = 0, \forall X$. We find that when using two learnable Lie algebra elements from the same input, the Lie bracket cancels out most of the information and only passes through the residual. As a result, we add a residual path to enhance the information flow, inspired by ResNet (He et al., 2016). The final design of the LN-Bracket layer becomes:

$$f_{\text{LN-Bracket}}(\mathbf{x}) = \mathbf{x} + [(\mathbf{x}U)^\wedge, (\mathbf{x}V)^\wedge]^\vee. \tag{15}$$

The nonlinear layer is often combined with a linear layer to form a module. In the rest of the paper, we will use LN-LR to denote an LN-Linear followed by an LN-ReLU, and LN-LB to denote an LN-Linear with an LN-Bracket layer.

### 4.3 POOLING LAYERS

Pooling layers provide a means to aggregate global information across the $N$ observation points within one measurement. This can be done by mean pooling, which is adjoint equivariant. In addition, we also introduce a max pooling layer. For each input $\mathcal{X} = \{\mathbf{x}_n\}_{n=1}^N \in \mathbb{R}^{K \times C \times N}$, and a weight matrix $\mathbf{W} \in \mathbb{R}^{C \times C}$, we learn a set of directions as: $\mathcal{D} = \{\mathbf{x}_n \mathbf{W}\}_{n=1}^N$.

We again employ the Killing form, $B(\cdot, \cdot)$, as the invariant function. For each channel $c \in C$, we have the max pooling function as $f_{\text{LN-Max}}(\mathbf{x}_n)[c] = \mathbf{x}_{n^*}[c]$, where

$$n^*(c) = \arg\max_n B(\mathbf{x}_n[c]\mathbf{W}, \mathbf{x}_n[c]). \tag{16}$$

$\mathbf{x}_n[c]$ denotes the input at channel $c$. Max pooling reduces the feature set from $\mathcal{X} \in \mathbb{R}^{K \times C \times N}$ to $\mathcal{X} \in \mathbb{R}^{K \times C \times 1}$. The layer is equivariant to the adjoint action due to the invariance of $B(\cdot, \cdot)$.

### 4.4 INVARIANT LAYERS

Equivariant layers allow steerable feature learning. However, some applications demand invariant features Lin et al. (2023); Zheng et al. (2022); Li et al. (2021). We introduce an invariant layer that can be attached to the network when necessary. Given an input $\mathbf{x} \in \mathbb{R}^{K \times C}$, we have:

$$f_{\text{LN-Inv}}(\mathbf{x}) = B(\mathbf{x}, \mathbf{x}), \tag{17}$$

where $B(\cdot, \cdot)$ is the adjoint-invariant Killing form, and $f_{\text{LN-Inv}}(\mathbf{x}) \in \mathbb{R}^{1 \times C}$.

### 4.5 RELATIONSHIP TO VECTOR NEURONS

Our method can almost specialize to the Vector Neurons when working with $\mathfrak{so}(3)$. This is because the linear adjoint matrix $Adm_a$ is exactly the rotation matrix for $\mathfrak{so}(3)$. Therefore, the group adjoint action becomes a left multiplication on the $\mathbb{R}^3$ representation of the Lie algebra. Moreover, SO(3) is a compact group. Thus, the negative Killing form of $\mathfrak{so}(3)$ defines an inner product. We omit the normalization in the ReLU layer because the norm is not well defined when the Killing form is not negative definite. We also do not implement a batch normalization layer. Therefore, the Lie Neurons do not reduce to VN completely when working with $\mathfrak{so}(3)$. Despite the similarity in appearance, the $\mathbb{R}^3$ vectors are viewed as vectors in Euclidean space subject to rotation matrix multiplication in Vector Neurons, while they are treated as $\mathfrak{so}(3)$ Lie algebras in our framework, where the rotation matrices stand for conjugation. In addition, we propose a novel Lie bracket nonlinear layer.

## 5 EXPERIMENTS

The Lie Neurons can be applied to any semisimple Lie algebra with a real matrix representation. This allows us to extend the network to operate on noncompact Lie algebras, such as the special

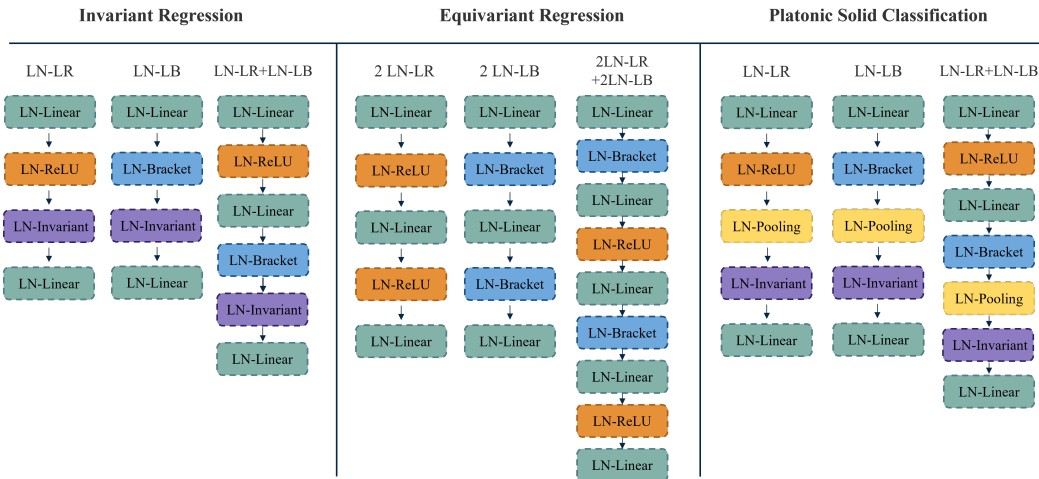

Figure 2: The network architecture used in each experiment.

linear Lie algebra $\mathfrak{sl}(3)$. In this section, we instantiate the LN on $\mathfrak{sl}(3)$, which can be represented using traceless matrices. The corresponding group, the special linear group $\mathrm{SL}(3)$, can be represented using matrices with unit determinants. The special linear group has 8 degrees of freedom and can be used to model the homography transformation between images Hua et al. (2020); Zhan et al. (2022).

We perform three different experiments to verify the LN. We first solve a regression problem on an invariant function, where the function maps two $\mathfrak{sl}(3)$ elements to a real number. In the second experiment, we fit an equivariant function that maps from $\mathfrak{sl}(3)$ to $\mathfrak{sl}(3)$. Lastly, we formulate a classification problem, where we classify among three Platonic solids. Across all three experiments, we compare our method with a standard 3-layer MLP by flattening the input to $\mathbb{R}^{K*C*N}$. In addition, we set the feature dimension to 256 for all models. The architecture of each model can be found in Figure 2.

## 5.1 INVARIANT FUNCTION REGRESSION

We begin our evaluation with an invariant function fitting experiment. Given $X, Y \in \mathfrak{sl}(3)$, we ask the network to regress the following function:

$$g(X, Y) = \sin(\mathrm{tr}(XY)) + \cos(\mathrm{tr}(YY)) - \frac{\mathrm{tr}(YY)^3}{2} + \det(XY) + \exp(\mathrm{tr}(XX)). \quad (18)$$

We randomly generate 10,000 training samples and 10,000 testing samples. In addition, in order to evaluate the invariance of the learned network, we randomly apply 500 group adjoint action to each test sample to generate augmented testing data.

In this task, we experiment with three different modules, `LN-LR`, `LN-LB`, and `LN-LR + LN-LB`, each followed by an `LN-Inv` and a final linear mapping from the feature dimension to a scalar. For each input, we concatenate $X$ and $Y$ in the feature dimension and have $\mathcal{X} \in \mathbb{R}^{K \times C \times N} = \mathbb{R}^{8 \times 2 \times 1}$. To the best of our knowledge, there is no existing adjoint-equivariant network. We additionally train the MLP with augmented data to serve as a stronger baseline.

To show the performance consistency, we train each model 5 separate times and calculate the mean and standard deviation of the performance. We report the Mean Squared Error (MSE) and the invariance error in Table 1. The invariance error $E_{\mathrm{inv}}$ is defined as:

$$E_{\mathrm{inv}} := \frac{\sum_{i=1}^{N_x} \sum_{j=1}^{N_a} f(\mathcal{X}_i) - f(a_j \mathcal{X}_i a_j^{-1})}{N_x N_a}, \quad (19)$$

where $a \in \mathrm{SL}(3)$ are the randomly generated adjoint actions, $N_x$ is the number of testing points, and $N_a$ is the number of conjugations. The invariance error measures the extent to which the model is actually invariant to the adjoint action.

Table 1: The mean squared errors and the invariant errors on the invariant function regression task. ↓ means the lower the better.

| Model | Training Augmentation | Num Params | Testing Augmentation | | | | Equivariance Error | |
| --- | --- | --- | --- | --- | --- | --- | --- | --- |
| | | | $Id$ | | $SL(3)$ | | | |
| | | | AVG ↓ | STD | AVG ↓ | STD | AVG ↓ | STD |
| MLP | $Id$ | 136,193 | 0.148 | 0.005 | 6.493 | 1.282 | 1.415 | 0.113 |
| MLP | $SL(3)$ | 136,193 | 0.201 | 0.01 | 1.119 | 0.018 | 0.683 | 0.006 |
| LN-LR | $Id$ | 66,562 | $1.30 \times 10^{-3}$ | $3.24 \times 10^{-5}$ | $1.30 \times 10^{-3}$ | $3.25 \times 10^{-5}$ | $3.60 \times 10^{-4}$ | $5.48 \times 10^{-5}$ |
| LN-LB | $Id$ | 132,098 | 0.557 | $1.87 \times 10^{-4}$ | 0.557 | $1.87 \times 10^{-4}$ | $\mathbf{1.43 \times 10^{-5}}$ | $1.42 \times 10^{-6}$ |
| LN-LR + LN-LB | $Id$ | 263,170 | $\mathbf{8.84 \times 10^{-4}}$ | $2.52 \times 10^{-5}$ | $\mathbf{8.84 \times 10^{-4}}$ | $2.49 \times 10^{-5}$ | $4.00 \times 10^{-4}$ | 0 |

From the table, we see that the LN outperform MLP except for `LN-LB`. When tested on the $SL(3)$ augmented test set, the performance of the LN remains consistent, while the error from the MLP increases significantly. The results of the invariance error demonstrate that the proposed method is invariant to the adjoint action while the MLP is not. Data augmentation helps MLP to perform better in the augmented test set, but at the cost of worse $Id$ test set performance, and the overall performance still lags behind our equivariant models. In this experiment, we observe that `LN-LR` performs well on the invariant task, but the `LN-LB` alone does not. Nevertheless, if we combine both nonlinearities, the performance remains competitive.

We additionally provide the training curves in Figure 3 to analyze the convergence property of the proposed network. We can see that the proposed method converges faster than the MLP, which indicates it is more data efficient. In addition, the MLP overfits to the training set and underperforms on the test set, while our method remains consistent.

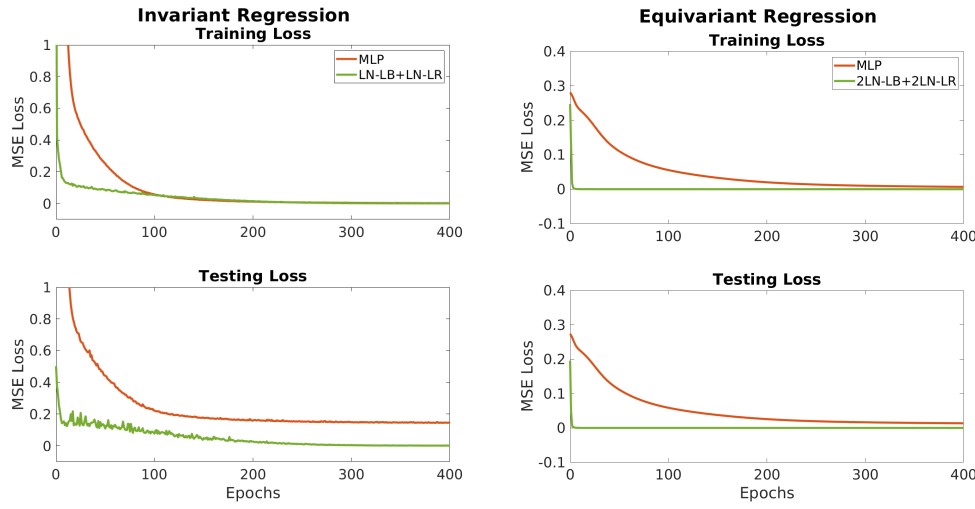

Figure 3: The training curves of the MLP and proposed method. We can see that in both invariant and equivariant regression tasks, our equivariant model converges much faster than MLPs, showing the data efficiency of our method.

## 5.2 EQUIVARIANT FUNCTION REGRESSION

In the second experiment, we ask the network to fit an equivariant function that takes two elements on $\mathfrak{sl}(3)$ back to itself:

$$h(X, Y) = [[X, Y], Y] + [Y, X]. \tag{20}$$

Similar to the first experiment, we generate $10,000$ training and test samples, as well as the additional 500 adjoint actions on the test set. For this task, we also train each model separately 5 times to analyze the consistency of proposed method. We again report the MSE on the regular test set. For the adjoint-augmented test set, we map the output back with the inverse adjoint action and compute

Table 2: The mean squared errors and the equivariant errors in equivariant function regression.

| Model | Training Augmentation | Num Params | Testing Augmentation | | | | Invariance Error | |
|---|---|---|---|---|---|---|---|---|
| | | | $Id$ | | SL(3) | | | |
| | | | AVG ↓ | STD | AVG ↓ | STD | AVG ↓ | STD |
| MLP | $Id$ | 538,120 | 0.011 | $3.53 \times 10^{-4}$ | 1.318 | $7.08 \times 10^{-2}$ | 0.424 | 0.003 |
| MLP | SL(3) | 538,120 | 0.033 | $2.86 \times 10^{-4}$ | 0.452 | $1.01 \times 10^{-2}$ | 0.389 | 0.001 |
| 2 LN-LR | $Id$ | 197,376 | 0.213 | $4.07 \times 10^{-5}$ | 0.213 | $4.08 \times 10^{-5}$ | $9.32 \times 10^{-5}$ | $6.65 \times 10^{-6}$ |
| 2 LN-LB | $Id$ | 328,448 | $\mathbf{9.83 \times 10^{-10}}$ | $1.78 \times 10^{-11}$ | $\mathbf{4.55 \times 10^{-8}}$ | $8.65 \times 10^{-11}$ | $\mathbf{6.56 \times 10^{-5}}$ | $4.22 \times 10^{-7}$ |
| 2 LN-LR + 2 LN-LB | $Id$ | 590,592 | $7.65 \times 10^{-9}$ | $3.54 \times 10^{-10}$ | $5.41 \times 10^{-8}$ | $4.08 \times 10^{-10}$ | $7.67 \times 10^{-5}$ | $1.56 \times 10^{-6}$ |

the MSE with the ground truth value. To evaluate the equivariance of the network, we compute the equivariance error $E_{\text{equiv}}$ as:

$$E_{\text{equiv}} := \frac{\sum_{i=1}^{N_x} \sum_{j=1}^{N_a} a_j f(\mathcal{X}_i) a_j^{-1} - f(a_j \mathcal{X}_i a_j^{-1})}{N_x N_a}. \tag{21}$$

In this experiment, we evaluate LN using 3 different architectures. They are 2 `LN-LR`, 2 `LN-LB`, and 2 `LN-LR` + 2 `LN-LB`, respectively. Each of them is followed by a regular linear layer to map the feature dimension back to 1.

Table 2 lists the results of the equivariant experiment. We see that the MLP performs well on the regular test set but fails to generalize to the augmented data. Moreover, it has a high equivariance error. Similar to the invariant task, data augmentation improves the MLP's performance on the augmented test set, but at the cost of worse $Id$ test set performance, and the overall performance still lags behind our equivariant models. Our methods, on the other hand, generalize well on the adjoint-augmented data and achieve the lowest errors. The 2 `LN-LB` model performs the best.

The training curves of this experiment is shown in Figure 3. The proposed network converges much faster than the MLP, which again demonstrates the data efficiency of the equivariant method.

From both the invariant and equivariant experiments, we observe that the `LN-LR` module works better on invariant tasks, while the `LN-LB` module performs better on the equivariant ones. We speculate this is because the `LN-LR` relies on the Killing form, which is an adjoint-invariant function, while the `LN-LB` leverages the Lie bracket, which is adjoint-equivariant. Nevertheless, if we combine both modules, the network performs favorably on both invariant and equivariant tasks.

## 5.3 PLATONIC SOLID CLASSIFICATION

Other than the numerical experiments above, we further design an experiment with practical meanings to hint at the real-world implications of the proposed network. The task is to classify polyhedrons from their projection on an image plane. While rotation equivariance naturally emerges for the 3D shape, the rotation equivariance relation is lost in the 2D projection of the 3D polyhedrons. Instead, the projection yields homography relations, which can be modeled using the SL(3) group Hua et al. (2020); Zhan et al. (2022). When projected onto an image plane, the two neighboring faces of a polyhedron can be described using homography transformations, which are different for each polyhedron type. Therefore, we use the homography transforms among the projected neighboring faces as the input for polyhedron classification.

Without loss of generality, we assume the camera intrinsic matrix $K$ to be identity. In this case, given a homography matrix $H \in \text{SL}(3)$ that maps one face to another in the image plane, the homography between these two faces becomes $RHR^{-1}$ when we rotate the camera by $R \in \text{SO}(3) \subset \text{SL}(3)$.

In this experiment, we use three types of Platonic solids: a tetrahedron, an octahedron, and an icosahedron. An input data point refers to the homography transforms between the projection of a pair of neighboring faces within one image. Figure 4 visualizes an example of the neighboring face pair for the three Platonic solids. The homographies of all neighboring face pairs form a complete set of data describing a Platonic solid. We use these data to learn a classification model of the three Platonic solids. During training, we fix the camera and object pose. Then, we test with the original pose and with rotated camera poses to verify the equivariance property of our models.

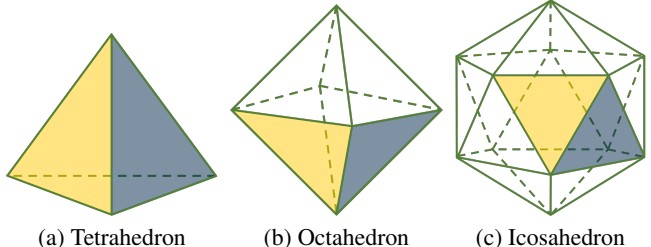

(a) Tetrahedron     (b) Octahedron     (c) Icosahedron

Figure 4: A visualization of the three Platonic solids in our classification task. The yellow and blue colors highlight a neighboring pair of faces, between which the homography transforms in the image plane are taken as input to our models.

Table 3: The accuracy of the Platonic solid classification task using the inter-face homography transforms in the image plane as inputs. ↑ means the higher, the better.

| Model | Num Params | Acc ↑ | | Acc (Rotated) ↑ | |
|---|---|---|---|---|---|
| | | AVG | STD | AVG | STD |
| MLP | 206,339 | 95.76% | 0.65% | 36.54% | 0.99% |
| LN-LR | 134,664 | 99.56% | 0.23% | 99.51% | 0.28% |
| LN-LB | 200,200 | 99.14% | 0.21% | 98.78% | 0.49% |
| LN-LR + LN-LB | 331,272 | **99.62**% | 0.25% | **99.61**% | 0.14% |

We once again test with `LN-LR`, `LN-LB`, and `LN-LR+LN-LB`. Each of them is followed by an `LN-Max` layer, an `LN-Inv` layer, and a final linear mapping. Each network is trained in 5 separate instances to analyze the consistency of the method.

Table 3 shows the classification accuracy. The LN achieves higher accuracy than the MLP. Since the MLP is not invariant to the adjoint action, its accuracy drops drastically when the camera is rotated. We also notice that the `LN-LB` performs slightly worse than the other two formulations. This agrees with our previous observations, as the classification tasks rely mostly on invariant features.

### 5.4 ABLATION STUDY

We introduce the `LN-Bracket` layer in Section 4.2.2 and discuss how the residual connection improves the performance. In this subsection, we perform ablation studies on an alternative Lie bracket nonlinear layer design without the residual connection. That is, $f_{\text{LN-Bracket-N}}(\mathbf{x}) = [(\mathbf{x}U)^{\wedge}, (\mathbf{x}V)^{\wedge}]^{\vee}$. We denote this nonlinear layer combined with an `LN-Linear` as `LN-LBN` and show the results of this method in Table 4. From the table, we can clearly see the benefits of having the residual connection in the Lie bracket layer.

### 5.5 POINT CLOUD CLASSIFICATION AND REGISTRATION

We also compare with Vector Neurons (Deng et al., 2021) in the $\mathfrak{so}(3)$ regime. Our method can be viewed as a generalization of Vector Neurons to semi-simple Lie algebras and mostly falls back to Vector Neurons for $\mathfrak{so}(3)$, except for the new Lie bracket nonlinear layer. We compare the performance of our network with the Lie bracket layer against the original Vector Neurons in the point cloud classification and registration tasks. The dataset is ModelNet40, containing object 3D models of 40 categories. The experimental setup follows (Deng et al., 2021) for classification and Zhu et al. (2022) for registration. The experimental results are shown in Table 5. As expected, our performance is very similar to the baseline. This demonstrates that the proposed network consistently generalizes Vector Neurons while maintaining the performance on SO(3) tasks.

## 6 DISCUSSION AND LIMITATIONS

Lie Neurons is a group adjoint equivariant network by construction. It does not require the Lie group to be compact. However, the `LN-ReLU` layer relies on a non-degenerated Killing form. As

Table 4: The ablation study of the Lie bracket layer in all three tasks. `LN-LBN` denotes the Lie bracket layer without the residual connection.

| | Invariant Regression | | | Equivariant Regression | | | Classification | |
|---|---|---|---|---|---|---|---|---|
| | MSE $\downarrow$ | MSE SL(3) $\downarrow$ | $E_{\text{inv}} \downarrow$ | MSE $\downarrow$ | MSE SL(3) $\downarrow$ | $E_{\text{equiv}} \downarrow$ | Acc $\uparrow$ | Acc (Rotated) $\uparrow$ |
| `LN-LB` | **0.558** | **0.558** | $4.9 \times 10^{-5}$ | $\mathbf{9.6 \times 10^{-10}}$ | $\mathbf{4.5 \times 10^{-8}}$ | $\mathbf{6.5 \times 10^{-5}}$ | **0.986** | **0.979** |
| `LN-LBN` | 4.838 | 4.838 | $\mathbf{2.4 \times 10^{-5}}$ | 0.276 | 0.276 | $2.7 \times 10^{-3}$ | 0.967 | 0.959 |

Table 5: Experimental results on point cloud classification and registration, testing SO(3) equivariance in comparison with Vector Neurons (Deng et al., 2021).

| Model | Point Cloud Classification (acc, % $\uparrow$) | Point Cloud Registration (angular error, deg $\downarrow$) | |
|---|---|---|---|
| | | Gaussian Noise $\sigma = 0.01$ | Uniform Noise $scale = 0.05$ |
| Vector Neurons | 73.70% | 2.23 | 1.63 |
| *Ours* | 73.50% | 2.53 | 1.13 |

a result, the current formulation can only operate on semisimple Lie algebras. Secondly, the Lie Neurons take elements on the Lie algebra as inputs, but most modern sensors return measurements in standard vector spaces. More practical applications of the proposed work are yet to be explored. Lastly, this work assumes a basis can be found for the target Lie algebra. However, for many applications such as robotics and computer vision, this assumption is valid.

## 7 CONCLUSION

In this paper, we propose an adjoint-equivariant network, Lie Neurons. Compared with existing equivariant models, our proposed framework extends the scope of equivariance by modeling the symmetry of change-of-basis on *transformations*, rather than vectors. Our model is generally applicable to any semisimple Lie groups, compact or non-compact. Our network builds upon simple MLP-style layers. To facilitate the learning of expressive Lie algebraic features, we propose equivariant nonlinear activation functions based on the Killing form and the Lie bracket. We also design an equivariant pooling layer and an invariant layer to extract global equivariant features and invariant features. In the experiments, we verify the equivariance property and the ability to fit equivariant and invariant targets of our model in regression and classification tasks, with $\mathfrak{sl}(3)$ as an example. We believe the new paradigm of transformation feature learning could open new possibilities in both equivariant modeling and more general deep learning.

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
