# OpenReview forum: "Lie Neurons: A General Adjoint-Equivariant Neural Network for Semisimple Lie Algebras"
_ICLR.cc/2024/Conference — Submitted to ICLR 2024_

### Official Review · Reviewer_mGDP · 2023-10-25

**Soundness:** 4 excellent
**Presentation:** 3 good
**Contribution:** 2 fair
**Rating:** 6
**Confidence:** 3

**Summary:**

This paper proposes Lie neurons, an equivariant neural architecture that accepts semisimple Lie group elements as inputs. Lie neurons can be seen as an extension of vector neurons, which consider SO(3) as data transformation. The authors handle mathematical tools such as Lie algebra, adjoint representation, etc to ensure the equivariance on more general class of groups. The proposed network is evaluated on three synthetic tasks.

**Strengths:**

The generalization of vector neurons from SO(3) to a semisimple Lie group is a novel contribution.

The paper is self-contained. It gently introduces the basics of Lie algebra necessary to understand the methodology.

**Weaknesses:**

Evaluation is the weakest point of this paper. There are three tasks: invariant regression, equivariant regression, and classification of polyhedrons. The first two tasks use synthetic data. Those are useful to know that the model is working as intended, but we cannot know the performance of real problems. The third task is more on the application side, but still synthetic and too clean (e.g. noise free).

Another concern is the range of applications. The problem setting --- the transformation itself is a data point --- is interesting, but it is not clear what kind of real problems can be formulated in that way. In other words, it's not straightforward to recognize the benefits of this framework from the practitioner's view.

**Questions:**

I don't wholly understand the setting of platonic solid classification.
1. The homography matrix H seems to be an input, but how is it computed from an image?
1. Also, the number of input points (the possible face pairs) varies depending on the platonic solids. How can you deal with the difference?
1. What is the network architecture used in this experiment?

How can we find $Adm$? For a given G (or its algebra g), is there any systematic way to calculate it?

---

> ### Author Response · Authors · 2023-11-23
> **Response to Reviewer mGDP (1)**
>
> **1. The problem of Lie algebra equivariance does not seem at all well-motivated by **applications** ... It is not clear in practice when one genuinely has input data that lives in a Lie algebra, rather than in the vector space associated with an ordinary group representation.**
>
> Thanks for pointing this out. The motivation is to provide a general equivariant framework for representation learning in problems with Lie group geometric structures. The Lie algebraic representation (adjoint) arises from standard group representation, making the proposed network as general as equivariance with respect to Lie group actions on vector spaces.
>
> We think the proposed network will have broad applications and has the potential to be adopted for applications such as 3D data processing using Inertial Measurement Units \[1,2] and Point Clouds \[3,4] where data can be viewed directly in $\mathfrak{so}(3)$ Lie algebra.
>
> There are also applications where the inputs are transformations of semi-simple groups, notably $SL(n)$. In computer vision, $SL(3)$ is the homography group \[5,6], which contains affine(2), and 2D affine transformations as its subgroup. Representation learning on $\mathfrak{sl}(3)$ will result in homography-invariant features applicable in numerous detection and classification tasks. For physical systems, $SL(3)$ is the volume-preserving transformation (a generalization of rigid body motion $SE(3)$ in kinematics and dynamics). We conjecture the proposed work can help predict the motion of deformable shapes, even though we do not have experimental results to back up this conjecture, and this is purely our geometric intuition from analytical mechanics.
>
> We envision our network to be applied in different applications in two ways:
>
>   1. For inputs that are directly transformations of semi-simple groups, we can directly apply our framework. This type of problems often arises in robotic state estimation and tracking, where the states being estimated are represented using elements of Lie groups. An example of this is homography tracking \[7], where measured homography matrices are noisy and fusion across multiple measurements is required.
>   2. If the measurements are raw sensor inputs which lives in some vector space, we need to find an adjoint-equivariant lift similar to Equation 13 in \[8]. This equivariant lift can be found heuristically, or it can be another equivariant neural network. For the special case where the input vectors are subjected to $SO(3)$ actions, such a lift will be identity. This is because the adjoint matrix of $SO(3)$ is exactly equal to the rotation matrix, i.e. $Adm\_R = R$, and $\mathfrak{so}(3)$ is isomorphic to $R^3$. We plan to explore a more general lifting network design in the near future.
>
> \[1] Hamel, Tarek, and Robert Mahony. "Attitude estimation on SO \[3] based on direct inertial measurements." _Proceedings 2006 IEEE International Conference on Robotics and Automation, 2006. ICRA 2006._. IEEE, 2006.
>
> \[2] Mahony, Robert, Tarek Hamel, and J-M. Pflimlin. "Complementary filter design on the special orthogonal group SO (3)." _Proceedings of the 44th IEEE Conference on Decision and Control_. IEEE, 2005.
>
> \[3] Deng, Congyue, et al. "Vector neurons: A general framework for so (3)-equivariant networks." _Proceedings of the IEEE/CVF International Conference on Computer Vision_. 2021.
>
> \[4] Zhu, Minghan, Maani Ghaffari, and Huei Peng. "Correspondence-free point cloud registration with SO (3)-equivariant implicit shape representations." _Conference on Robot Learning_. PMLR, 2022.
>
> \[5] Manerikar, Ninad, Minh-Duc Hua, and Tarek Hamel. "Homography observer design on special linear group SL (3) with application to optical flow estimation." _2018 European Control Conference (ECC)_. IEEE, 2018.
>
> \[6] Mei, Christopher, et al. "Homography-based tracking for central catadioptric cameras." _2006 IEEE/RSJ International Conference on Intelligent Robots and Systems_. IEEE, 2006.
>
> \[7] Hamel, Tarek, et al. "Homography estimation on the special linear group based on direct point correspondence." _2011 50th IEEE Conference on Decision and Control and European Control Conference_. IEEE, 2011.
>
> \[8] van Goor, Pieter, Tarek Hamel, and Robert Mahony. "Equivariant filter (eqf)." _IEEE Transactions on Automatic Control_ (2022).

---

> ### Author Response · Authors · 2023-11-23
> **Response to Reviewer mGDP (2)**
>
> **2. Evaluation is the weakest point of this paper. There are three tasks: invariant regression, equivariant regression, and classification of polyhedrons. The first two tasks use synthetic data. Those are useful to know that the model is working as intended, but we cannot know the performance of real problems. The third task is more on the application side, but still synthetic and too clean (e.g. noise free).**
>
> Thank you for the comments. We have identify several potential applications in question 1. In order to show the immediate connection with practical applications, we compare with Vector Neurons in the $\mathfrak{so}(3)$ regime. Our method can be viewed as a generalization of Vector Neurons to semi-simple Lie algebras, and mostly falls back to Vector Neurons for $\mathfrak{so}(3)$, except for the new Lie bracket nonlinear layer. We compare the performance of our network with the Lie bracket layer against the original Vector Neurons, in the point cloud classification \[3] and registration \[4] tasks. As expected, our performance is very similar to the baseline. This demonstrates that the proposed network generalizes Vector Neurons, while maintaining the performance on $SO(3)$ tasks.
>
> |                |                            |                          |               |
> | -------------- | -------------------------: | -----------------------: | ------------: |
> |                | Point Cloud Classification | Point Cloud Registration |               |
> |                |                            |           Gaussian Noise ($\sigma=0.01$) | Uniform Noise ($scale=0.05$) |
> | Vector Neurons |                     73.70% |                   2.2263 |        1.6305 |
> | Ours           |                     73.50% |                   2.5343 |        1.1324 |
>
> \[3] Deng, Congyue, et al. "Vector neurons: A general framework for so (3)-equivariant networks." _Proceedings of the IEEE/CVF International Conference on Computer Vision_. 2021.
>
> \[4] Zhu, Minghan, Maani Ghaffari, and Huei Peng. "Correspondence-free point cloud registration with SO (3)-equivariant implicit shape representations." _Conference on Robot Learning_. PMLR, 2022.
>
> **3. I don't wholly understand the setting of platonic solid classification.**
>
> **3.1. The homography matrix H seems to be an input, but how is it computed from an image?**
>
> In the platonic solids investigated in this paper, each face has three vertices. Thus, for each pair of neighboring faces, there are three possible homographies to map between them in the projected image, each assuming a different correspondence between the three vertices of the two faces by rotating the vertices. The three homographies form the input $\mathfrak{sl}(3)$ features for this pair of neighboring faces. For the classification task, the network samples a fixed number of pairs of neighboring faces to form the input feature of this platonic solid. Such homography features are subject to adjoint actions when the camera rotates, to which our model is equivariant.
>
> **3.2. Also, the number of input points (the possible face pairs) varies depending on the platonic solids. How can you deal with the difference?**
>
> As mentioned above, a fixed number (12) of pair-wise homographies are sampled from each platonic solid.
>
> **3.3. What is the network architecture used in this experiment?**
>
> Thank you for pointing this out. We have added an illustration of the architecture being used in each experiment in Figure 2.
>
> **4. How can we find Adm? For a given G (or its algebra g), is there any systematic way to calculate it?**
>
> Yes, there is an systematic way to calculate the Adjoint matrix. In this work, we focuses on matrix Lie groups. Since Lie algebra is a vector space, we can always find a basis. With this basis, each element in the Lie algebra can be represented using a $R^n$ vector. The adjoint matrix $Adm$ that maps from an vector $v_1 \in R^n$ to $v_2 \in R^n$ can be computed following the procedure described in Chapter 10.5.2 in [9].
>
> \[9] Chirikjian, Gregory S. Stochastic models, information theory, and Lie groups, volume 2: Analytic methods and modern applications. Vol. 2. Springer Science & Business Media, 2011.

---

### Official Review · Reviewer_Wrrb · 2023-10-31

**Soundness:** 2 fair
**Presentation:** 2 fair
**Contribution:** 2 fair
**Rating:** 5
**Confidence:** 3

**Summary:**

Inspired by the work on vector neurons, this paper proposes neural network layers which capture equivariance to the adjoint action of a Lie group on its Lie algebra, that is, action by conjugation. Such a network would take elements of the Lie algebra as input. The novel layers include a linear layer, two different nonlinearities, and a pooling layer. The nonlinearities draw on the notion of the Killing form, requiring it to be negative definite and thus restricting these particular layers to semisimple Lie algebras. The paper then runs three different synthetic experiments, one learning an adjoint action-invariant function, one learning an adjoint action-equivariant function, and one taking as input transformations between faces of 2-dimensional projections of 3-dimensional solids and trying to predict the solid. The paper compares networks with various combinations of their proposed layers with vanilla MLPs.

**Strengths:**

**Interesting ideas and interesting math:** The ideas contained in this paper are interesting, particularly envisioning neural networks as functions on Lie algebras. The reviewer appreciated central elements (such as the Killing form) from Lie theory cleverly integrated into neural network design. Most of the mathematical constructions seem relatively well-thought out (though this reviewer did not check the equivariance or invariance on each). Further, the work appears to be a nice generalization of a prior approach.

**Good pacing and background:** Papers on equivariance in deep learning can be difficult to read since they draw on two different technical domains and the conversion of constructions and notation from representation theory to deep learning is fraught with peril. While the reviewer believes that some aspects of the writing deserve further work, the pacing (not too fast) and amount of background (enough for someone with sufficient mathematical maturity to understand) stood out as strengths. Because of this, the work was enjoyable to read.

**Weaknesses:**

**Motivation:** From a purely mathematical perspective, the problem is interesting. However, this reviewer’s estimation is that to have a substantial impact, equivariant research needs to have some application to real data. This paper does not really have any experiments on real data, nor did this reviewer see a clear path towards an application. The Platonic solids experiment seemed to be aimed at real applications, but it was a little unclear to this reviewer what these would be. It would be helpful to outline how this type of problem arises in the real world even if no experiments are run with real data (in particular, if homography modeling is a real problem that one can imagine this network architecture being applied to, perhaps more details could be given).

**Limited experiments:** This reviewer has two concerns about the experimental section. The first was the lack of analysis and the second was the lack of strong baselines and experimental breadth. As this is a novel architecture, it would be useful to see analysis of model performance beyond a single metric. For example, training curves and convergence, confusion matrices, etc. This would allow the reader to better understand what the model is and is not actually good at. For instance, empirically, equivariant models are often more data efficient, is that the case here? It might also make sense to measure the extent to which the proposed model is actually invariant to the action that it is claimed to be invariant to. Of course, no paper can run all possible experiments, but some additional experiments would be useful. This reviewer felt that they did not have a strong feel for the method, despite reading through three different experiments.

It would also improve the work if stronger baselines were compared against. For instance, what happens if we try to use some type of adjoint-action augmentation when training the vanilla MLP? It would also be interesting to see how the model performs with different Lie algebras, are some Lie algebras harder to learn on than overs? How does performance scale with Lie algebra size?

**Writing and figures:** While the reviewer certain aspects of the writing, there are other ways that the the paper could be made more clear.
- **Moving between elements of the Lie algebra and corresponding vectors for processing by the model:** One of the aspects of the paper most likely to cause the reader confusion is the process of moving between the Lie algebra $\mathfrak{g}$ itself and the corresponding vector space $\mathbb{R}^m$. As is pointed out in the Nitpicks section, one problem is that $E_i$ is never defined other than saying it is the image of $e_i$ under the map $\wedge$. But then the definition of $\wedge$ uses $E_i$. More broadly, it might be worth investigating sweeping the distinction between the Lie algebra and its underlying vector space under the rug to cut down on extra notation.
- **Figure 1:** This reviewer found Figure 1 hard to understand. It might be helpful to somehow include the model in the schematic so it is more clear what is the input/output, group action on the input vs group action on the output, etc.

### Nitpicks

- $E_i$ is never defined beyond $E_i = (e_i)^\vee \in \mathfrak{g}$. It would be helpful to say something more explicit.
- $\log$ isn’t actually used in the paper? Perhaps (2) could be removed.
- For equalities in (13), it might make sense to state the reason all this works (the matrix $Adm_a$ and $W$ are being multiplied from different sides?
- The experimental results should have confidence intervals attached to them.

**Questions:**

The reviewer asked a number of questions in the prior sections. These include:
- Thoughts on how the proposed architecture would scale, both to higher dimensional Lie algebras and larger volumes of data.
- More details on convergence, data efficiency, and other auxiliary metrics that can give a better picture of these models.
- Discussion of what types of real data this architecture might be applied to.

---

> ### Author Response · Authors · 2023-11-23
> **Response to Reviewer Wrrb (1)**
>
> **1. Motivation: From a purely mathematical perspective, the problem is interesting. However, this reviewer’s estimation is that to have a substantial impact, equivariant research needs to have some application to real data. This paper does not really have any experiments on real data, nor did this reviewer see a clear path towards an application. The Platonic solids experiment seemed to be aimed at real applications, but it was a little unclear to this reviewer what these would be. It would be helpful to outline how this type of problem arises in the real world even if no experiments are run with real data (in particular, if homography modeling is a real problem that one can imagine this network architecture being applied to, perhaps more details could be given).**
>
> We would like to thank the reviewer for the comment. The motivation is to provide a general equivariant framework for representation learning in problems with Lie group geometric structures. The Lie algebraic representation (with adjoint group action) arises from standard group representation, making the proposed network as general as equivariance with respect to Lie group actions on vector spaces.
>
> We think the proposed network will have broad applications and has the potential to be adopted for applications such as 3D data processing using Inertial Measurement Units \[1,2] and Point Clouds \[3,4] where data can be viewed directly in $\mathfrak{so}(3)$ Lie algebra.
>
> There are applications where the inputs are transformations of semi-simple groups, notably $SL(n)$. In computer vision, $SL(3)$ is the homography group \[5,6], which contains affine(2), and 2D affine transformations as its subgroup. Representation learning on $\mathfrak{sl}(3)$ will result in homography-invariant features applicable in numerous detection and classification tasks. For physical systems, $SL(3)$ is the volume-preserving transformation (a generalization of rigid body motion $SE(3)$ in kinematics and dynamics). We conjecture the proposed work can help predict the motion of deformable shapes, even though we do not have experimental results to back up this conjecture, and this is purely our geometric intuition from analytical mechanics.
>
>  We envision our network to be applied in different applications in two ways:
>   1. For inputs that are directly transformations of semi-simple groups, we can directly apply our framework. This type of problem often arises in robotic state estimation and tracking, where the states being estimated are represented using elements of Lie groups. An example of this is homography tracking \[7], where measured homography matrices are noisy and fusion across multiple measurements is required.
>   2. If the measurements are raw sensor inputs which lives in some vector space, we need to find an adjoint-equivariant lift similar to Equation 13 in \[8]. This equivariant lift can be found heuristically, or it can be another equivariant neural network. For the special case where the input vectors are subjected to $SO(3)$ actions, such a lift will be the identity map. This is because the adjoint matrix of $SO(3)$ is exactly equal to the rotation matrix, i.e. $Adm\_R = R$, and $\mathfrak{so}(3)$ is isomorphic to $R^3$. To be clear, a lift from the input to the symmetry group is always possiple, e.g., via sampling, but the challenge is in constructing an equivariant lift. We plan to explore a more general lifting network design in the near future.
>
> \[1] Hamel, Tarek, and Robert Mahony. "Attitude estimation on SO \[3] based on direct inertial measurements." _Proceedings 2006 IEEE International Conference on Robotics and Automation, 2006. ICRA 2006._. IEEE, 2006.
>
> \[2] Mahony, Robert, Tarek Hamel, and J-M. Pflimlin. "Complementary filter design on the special orthogonal group SO (3)." _Proceedings of the 44th IEEE Conference on Decision and Control_. IEEE, 2005.
>
> \[3] Deng, Congyue, et al. "Vector neurons: A general framework for so (3)-equivariant networks." _Proceedings of the IEEE/CVF International Conference on Computer Vision_. 2021.
>
> \[4] Zhu, Minghan, Maani Ghaffari, and Huei Peng. "Correspondence-free point cloud registration with SO (3)-equivariant implicit shape representations." _Conference on Robot Learning_. PMLR, 2022.
>
> \[5] Manerikar, Ninad, Minh-Duc Hua, and Tarek Hamel. "Homography observer design on special linear group SL (3) with application to optical flow estimation." _2018 European Control Conference (ECC)_. IEEE, 2018.
>
> \[6] Mei, Christopher, et al. "Homography-based tracking for central catadioptric cameras." _2006 IEEE/RSJ International Conference on Intelligent Robots and Systems_. IEEE, 2006.
>
> \[7] Hamel, Tarek, et al. "Homography estimation on the special linear group based on direct point correspondence." _2011 50th IEEE Conference on Decision and Control and European Control Conference_. IEEE, 2011.
>
> \[8] van Goor, Pieter, Tarek Hamel, and Robert Mahony. "Equivariant filter (eqf)." _IEEE Transactions on Automatic Control_ (2022).

---

> > ### Author Response · Authors · 2023-11-23
> > **Response to Reviewer Wrrb (2)**
> >
> > **2. Limited experiments: This reviewer has two concerns about the experimental section. The first was the lack of analysis and the second was the lack of strong baselines and experimental breadth.**
> >
> >   **a. As this is a novel architecture, it would be useful to see an analysis of model performance beyond a single metric. For example, training curves and convergence, confusion matrices, etc. This would allow the reader to better understand what the model is and is not actually good at. For instance, empirically, equivariant models are often more data efficient, is that the case here? It might also make sense to measure the extent to which the proposed model is actually invariant to the action that it is claimed to be invariant to. Of course, no paper can run all possible experiments, but some additional experiments would be useful. This reviewer felt that they did not have a strong feel for the method, despite reading through three different experiments.**
> >
> > Thank you for your valuable comments. We conducted extra experiments as suggested by the reviewers to provide extra information to demonstrate the properties of our method.
> >
> > First, in terms of data efficiency, we present the training curves of our method and the baseline MLP in Figure 3. We can see that in both invariant and equivariant regression tasks, our equivariant model converges much faster than MLPs, showing the data efficiency of our method.
> >
> > Second, in terms of the invariant property, it is shown through the invariance/equivariance error in our paper, which indicates the output difference when input undergoes the adjoint action. Our models present almost zero invariance/equivariance error, indicating that our model is exactly equivariant. Our models have identical performance between Id and SL(3) test sets, supporting that our model is exactly equivariant.
> >
> > Third, in terms of the confusion matrix, we did not include it in our paper because the accuracy numbers show most information in the platonic solids classification task. Near 100% accuracy shows that our model performs almost perfectly in this task, while the 36% accuracy shows that the MLP is as good as random guesses when the data is augmented by camera rotation.
> > Furthermore, to show the performance consistency, we trained each model 5 separate times, and calculated the mean and standard deviation of the performance, as shown in the tables below. Our networks present very small std values in the performance, indicating that the high performance of our networks is reliable and repeatable.
> > |               |                       |                    |                   |          |              |          |                  |          |
> > | :-----------: | :-------------------: | :----------------: | :---------------: | :------: | :----------: | :------: | :--------------: | :------: |
> > |               |                       | **Invariant Task** |                   |          |              |          |                  |          |
> > |     Model     | Training augmentation |     Num Params     | Test augmentation |          |              |          | Invariance Error |          |
> > |               |                       |                    |         Id        |          |     SL(3)    |          |                  |          |
> > |               |                       |                    |        AVG        |    STD   |      AVG     |    STD   |        AVG       |    STD   |
> > |      MLP      |           Id          |       136,193      |       0.148       |   0.005  |     6.493    |   1.282  |       1.415      |   0.113  |
> > |      MLP      |         SL(3)         |       136,193      |       0.201       |   0.010  |     1.119    |   0.018  |       0.683      |   0.006  |
> > |     LN-LR     |           Id          |       66,562       |      1.30E-03     | 3.24E-05 |   1.30E-03   | 3.25E-05 |     3.60E-04     | 5.48E-05 |
> > |     LN-LB     |           Id          |       132,098      |       0.557       | 1.87E-04 |     0.557    | 1.87E-04 |   **1.43E-05**   | 1.42E-06 |
> > | LN-LR + LN-LB |           Id          |       263,170      |    **8.84E-04**   | 2.52E-05 | **8.84E-04** | 2.49E-05 |     4.00E-04     |   0.000  |

---

> ### Author Response · Authors · 2023-11-23
> **Response to Reviewer Wrrb (3)**
>
> |         |      |             |       |          |        |     |          |    |
> | :---------------: | :-------------------: | :------------------: | :---------------: | :------: | :----------: | :------: | :----------------: | :------: |
> |          |         | **Equivariant Task** |          |          |         |          |           |          |
> |       Model       | Training augmentation |      Num Params      | Test augmentation |          |         |        | Equivariance Error |          |
> |      |       |      |         Id        |          |     SL(3)    |          |         |          |
> |      |       |      |        AVG        |    STD   |      AVG     |    STD   |         AVG        |    STD   |
> |        MLP        |           Id          |        538,120       |       0.011       | 3.53E-04 |     1.318    | 7.08E-02 |        0.424       |   0.003  |
> |        MLP        |         SL(3)         |        538,120       |       0.033       | 2.86E-04 |     0.452    | 1.01E-02 |        0.389       |   0.001  |
> |      2 LN-LR      |           Id          |        197,376       |       0.213       | 4.07E-05 |     0.213    | 4.08E-05 |      9.32E-05      | 6.65E-06 |
> |      2 LN-LB      |           Id          |        328,448       |    **9.83E-10**   | 1.78E-11 | **4.55E-08** | 8.65E-11 |    **6.56E-05**    | 4.22E-07 |
> | 2 LN-LR + 2 LN-LB |           Id          |        590,592       |      7.65E-09     | 3.54E-10 |   5.41E-08   | 4.08E-10 |      7.67E-05      | 1.56E-06 |
>
> |      |      |      |     |      |     |
> | :-----------: | :---------------------: | :----: | :---: | :-----------: | :---: |
> |            | **Classification Task** |        |       |               |       |
> |            |        Num Params       |   Acc  |       | Acc (Rotated) |       |
> |            |                         |   AVG  |  STD  |      AVG      |  STD  |
> |      MLP      |         206,339         | 95.76% | 0.65% |     36.54%    | 0.99% |
> |     LN-LR     |         134,664         | 99.56% | 0.23% |     99.51%    | 0.28% |
> |     LN-LB     |         200,200         | 99.14% | 0.21% |     98.78%    | 0.49% |
> | LN-LR + LN-LB |         331,272         | **99.62%** | 0.25% |    **99.61%**    | 0.14% |
>
> **b. It would also improve the work if stronger baselines were compared against. For instance, what happens if we try to use some type of adjoint-action augmentation when training the vanilla MLP?**
>
> We agree that we can run MLP with adjoint data augmentation for comparison. We have conducted new experiments on this setup. The results are shown in the tables above. MLP with $SL(3)$ adjoint augmentation during training still largely underperforms our equivariant models.
>
> **c. It would also be interesting to see how the model performs with different Lie algebras, are some Lie algebras harder to learn on than others? How does performance scale with Lie algebra size?**
>
> The model size is independent of the Lie algebra size since the dimension corresponding to the Lie algebra is kept throughout the layers. In terms of the performance, it is tricky to directly compare across different Lie algebras with a consistent metric, but we do not see a scaling problem for the network to process Lie algrbras of different sizes.
> To show the performance on other Lie algebras, we added experiments on $\mathfrak{so}(3)$. Our method can be viewed as a generalization of Vector Neurons to semi-simple Lie algebras, and mostly falls back to Vector Neurons for $\mathfrak{so}(3)$, except for the new Lie bracket nonlinear layer. We compare the performance of our network with the Lie bracket layer against the original Vector Neurons, in the point cloud classification [3] and registration [4] tasks. As expected, our performance is very similar to the baseline. This demonstrates that the proposed network generalizes Vector Neurons, while maintaining the performance on $SO(3)$ tasks.
>
> ****
>
> |                |                            |                          |               |
> | -------------- | -------------------------: | -----------------------: | ------------: |
> |                | Point Cloud Classification (acc, \% ) |  |Point Cloud Registration      (angular error, deg)          |
> |                |                                          |           Gaussian Noise $\sigma=0.01$  | Uniform Noise $scale=0.05$|
> | Vector Neurons |                     73.70% |                   2.2263 |        1.6305 |
> | Ours           |                     73.50% |                   2.5343 |        1.1324 |
>
> ****
> [3] Deng, Congyue, et al. "Vector neurons: A general framework for so (3)-equivariant networks." Proceedings of the IEEE/CVF International Conference on Computer Vision. 2021.
>
> [4] Zhu, Minghan, Maani Ghaffari, and Huei Peng. "Correspondence-free point cloud registration with SO (3)-equivariant implicit shape representations." Conference on Robot Learning. PMLR, 2022.

---

> ### Author Response · Authors · 2023-11-23
> **Response to Reviewer Wrrb (4)**
>
> **3. Writing and figures: While the reviewer certain aspects of the writing, there are other ways that the the paper could be made more clear:**
>
>   **a. Moving between elements of the Lie algebra and corresponding vectors for processing by the model: One of the aspects of the paper most likely to cause the reader confusion is the process of moving between the Lie algebra g itself and the corresponding vector space R^m. As is pointed out in the Nitpicks section, one problem is that E_i is never defined other than saying it is the image of e_i under the map ∧. But then the definition of ∧ uses E_i. More broadly, it might be worth investigating sweeping the distinction between the Lie algebra and its underlying vector space under the rug to cut down on extra notation.**
>
> We apologize for the confusion. In this work, we focus on finite-dimensional matrix Lie algebras. The wedge is an operation that generates the Lie algebra elements using the canonical basis. Because Lie algebra is a vector space, such a basis always exists. For example, the $\mathfrak{so}(3)$ elements has the form $\begin{bmatrix}0& -\\omega_z & \\omega_y \\\ \\omega_z & 0 & -\\omega_x \\\ -\\omega_y & \\omega_x & 0 \\end{bmatrix}$. One can find the canonical basis as $E_x = \begin{bmatrix} 0& 0 & 0\\\ 0&0&-1\\\ 0&1&0 \end{bmatrix}, E_y = \\begin{bmatrix} 0& 0 & 1\\\ 0&0&0\\\ -1&0&0 \\end{bmatrix}, E_z = \begin{bmatrix} 0& -1 & 0 \\\ 1&0&0\\\ 0&0&0 \end{bmatrix}$.
> With $v=\begin{bmatrix} \omega_x, \omega_y, \omega_z \end{bmatrix}$, we have $(v)^\wedge = \omega_x E_x + \omega_y E_y + \omega_z E_z = \begin{bmatrix}0& -\\omega_z & \\omega_y \\\ \\omega_z & 0 & -\\omega_x \\\ -\\omega_y & \\omega_x & 0 \\end{bmatrix} = W \in \mathfrak{so}(3)$.
> The Vee operation is the inverse of this: $(W)^\vee = v$
> We will update the paper in Section 3.1 with this example to better convey the concept.
> For a detailed description, we kindly refer the reviewer to Equation 10.30 and Chapter 10.2.4 in [9]. In our experiments, we use the basis for $\mathfrak{sl}(3)$ from [10].
>
> [9] Chirikjian, Gregory S. Stochastic models, information theory, and Lie groups, volume 2: Analytic methods and modern applications. Vol. 2. Springer Science & Business Media, 2011.
>
> [10] Winternitz, Pavel. "Subalgebras of Lie algebras. Example of sl (3, R)." Centre de Recherches Mathématiques CRM Proceedings and Lecture Notes. Vol. 34. 2004.
>
>   **b. Figure 1: This reviewer found Figure 1 hard to understand. It might be helpful to somehow include the model in the schematic so it is more clear what is the input/output, group action on the input vs group action on the output, etc.**
>
> We thank the reviewers for the valuable suggestions. We have added a model schematic with annotations on input/output equivariant relationship in Figure 1.
>
> **4. E\_i is never defined beyond E\_i=(e\_i)∨∈g. It would be helpful to say something more explicit.**
>
> We agree with the reviewer. In the experiments, we use the canonical basis defined in \[10]. We will cite this reference in the revised paper.
>
>    \[10] Winternitz, Pavel. "Subalgebras of Lie algebras. Example of sl (3, R)." _Centre de Recherches Mathématiques CRM Proceedings and Lecture Notes_. Vol. 34. 2004.
>
> **5. Log isn’t actually used in the paper? Perhaps (2) could be removed.**
>
> We agree with the reviewer that the Log is not needed in our context. We will remove it from the paper.
>
> **6. For equalities in (13), it might make sense to state the reason all this works (the matrix Adm and w are being multiplied from different sides?**
>
> We inherit the design from Vector Neurons. The feature is of shape K\*C, where K is the dimension of the Lie algebra (or the dimension of the Euclidean space for Vector Neurons), and C is the dimension of the feature channels. A linear layer can be viewed as a matrix multiplication on the right (C dimension), while the adjoint matrix acts on the left (K dimension). We thank the reviewer for the suggestion. We will add the above explanation in Section 4.1 in the revised paper.
>
> **7. The experimental results should have confidence intervals attached to them.**
>
> Thank you for pointing this out. We have conducted new experiments to address this issue. Specifically, we train each network separately 5 times and compute the average and standard deviation over the same model trained in different instances. The result is shown in Table 1 - 3.

---

> ### Author Response · Authors · 2023-11-23
> **Response to Reviewer Wrrb (5)**
>
> Questions:
>
> **1. Thoughts on how the proposed architecture would scale, both to higher dimensional Lie algebras and larger volumes of data.**
>
> We have provided our response in question 2. We kindly ask the reviewer to refer to it.
>
> **2. More details on convergence, data efficiency, and other auxiliary metrics that can give a better picture of these models.**
>
> We have conducted several new experiments and provided extra informations based on the reviewer’s suggestions. Please refer to question 2 for detail descriptions.
>
> **3. Discussion of what types of real data this architecture might be applied to.**
>
> We will improve the motivation as discussed in question 1 in the paper.

---

### Official Review · Reviewer_272k · 2023-11-01

**Soundness:** 3 good
**Presentation:** 2 fair
**Contribution:** 2 fair
**Rating:** 5
**Confidence:** 3

**Summary:**

The authors build an architecture which takes as input elements in the Lie algebra vector space of a Lie group, and is equivariant to the adjoint representation of the Lie algebra. As building blocks, they use nonlinearities based on the Killing form and the Lie bracket, as well as ordinary linear layers. Their architecture generalizes vector neurons beyond the Lie algebra of SO(3). They obtain higher test accuracies than ordinary MLPs on three synthetic tasks, including a homography transform task on Platonic solids.

**Strengths:**

This paper is the first to prescribe Lie algebra equivariance for a neural network. It applies to all semi simple Lie groups’ algebras, which is general, and the resultant architecture is exactly equivariant. The use of the Killing form in a neural architecture seems to be novel, as is the explicit use of the Lie bracket. The experimental results are better than the baseline MLPs.

**Weaknesses:**

1. The problem of Lie algebra equivariance does not seem at all well-motivated by applications. This is evidenced by the datasets used in experiments: the first two are quite contrived, and the third one, although geometric in nature, remains synthetic. It is not clear in practice when one genuinely has input data that lives in a Lie algebra, rather than in the vector space associated with an ordinary group representation.
2. There is no end-to-end description of the architecture. Also, the architecture is not motivated in a particularly natural or pedagogical way, e.g. with a discussion of universality, a comparison to other approaches one might take for linear equivariance, etc. Generally speaking, the presentation could be improved — to include more details, to better motivate the equivariance problem via applications, and conditioned on this setting, to better motivate this particular architecture.
3. The only baseline included in experiments is an MLP. It would make sense to include an architecture equivariant/invariant to all linear transformations, for example, or an MLP augmented with Lie algebra transformations. On its own, the MLP seems like a weak baseline.

As a minor comment, I found the analogy made to vector neurons more confusing than enlightening. It would be helpful to include a self-contained description of vector neurons, to make this comparison more clear. But overall, vector neurons are just one example of an equivariant architecture whose input representations transform according to rotations. Another important feature of vector neurons is that one representation is maintained for each point in the input point cloud, but this does not appear to have an analogue here (please correct me if I am misunderstanding).

**Questions:**

1. What is the wedge notation ^ used in equation 4? Is this defined anywhere?
2. I don’t understand equation 10, defining the Killing form. If I understand correctly, $ad_X$ is a function from the Lie algebra to itself, sending an input M to [X, M] = XM - MX. $ad_X \circ ad_Y$ is then also a function from the Lie algebra to itself. What is the trace of this function? Should we imagine setting a basis for the finite-dimensional Lie algebra, constructing the matrix that represents $ad_X \circ ad_Y$ based on this basis, and computing the trace (which I imagine would actually be invariant to the choice of basis)?
3. Is the proposed architecture universal over continuous Lie algebra equivariant functions?
4. The authors refer to the input data as “transformations”, but also as elements of the Lie algebra. Why is it helpful to conceptualize elements of the Lie algebra as transformations? Does this interpretation still make sense for Lie groups which are not matrix Lie groups?
5. How many parameters are in the baseline MLP, compared to the LN-LR and LN-LB architectures in the experiments?
6. As far as I understand, the adjoint operation is a particular subset of general linear transformations of the inputs. However, the linear layer is equivariant to any linear transformation, not just the adjoint operation, correct? So to clarify, are the Killing form and the Lie bracket operations special nonlinearities, in that they are only equivariant to the adjoint operation (but not to more general linear transformations)? It looks to me like the Lie bracket nonlinearity in equation (16) would be invariant general unitary transformations, but I do not understand the wedge notation, so this may not be the case.
7. Why is the adjoint operation meaningful to study? Does it arise often in applications? And can it not be expressed as a subgroup of GL(d), where d is the dimension of the Lie algebra?

---

> ### Author Response · Authors · 2023-11-23
> **Response to Reviewer 272k (1)**
>
> **1. The problem of Lie algebra equivariance does not seem at all well-motivated by **applications** ... It is not clear in practice when one genuinely has input data that lives in a Lie algebra, rather than in the vector space associated with an ordinary group representation.**
>
> Thanks for pointing this out. The motivation is to provide a general equivariant framework for representation learning in problems with Lie group geometric structures. The Lie algebraic representation (adjoint) arises from standard group representation, making the proposed network as general as equivariance with respect to Lie group actions on vector spaces.
>
> We think the proposed network will have broad applications and has the potential to be adopted for applications such as 3D data processing using Inertial Measurement Units \[1,2] and Point Clouds \[3,4] where data can be viewed directly in $\mathfrak{so}(3)$ Lie algebra.
>
> There are also applications where the inputs are transformations of semi-simple groups, notably $SL(n)$. In computer vision, $SL(3)$ is the homography group \[5,6], which contains affine(2), and 2D affine transformations as its subgroup. Representation learning on $\mathfrak{sl}(3)$ will result in homography-invariant features applicable in numerous detection and classification tasks. For physical systems, $SL(3)$ is the volume-preserving transformation (a generalization of rigid body motion $SE(3)$ in kinematics and dynamics). We conjecture the proposed work can help predict the motion of deformable shapes, even though we do not have experimental results to back up this conjecture, and this is purely our geometric intuition from analytical mechanics.
>
> We envision our network to be applied in different applications in two ways:
>
>   1. For inputs that are directly transformations of semi-simple groups, we can directly apply our framework. This type of problems often arises in robotic state estimation and tracking, where the states being estimated are represented using elements of Lie groups. An example of this is homography tracking \[7], where measured homography matrices are noisy and fusion across multiple measurements is required.
>   2. If the measurements are raw sensor inputs which lives in some vector space, we need to find an adjoint-equivariant lift similar to Equation 13 in \[8]. This equivariant lift can be found heuristically, or it can be another equivariant neural network. For the special case where the input vectors are subjected to $SO(3)$ actions, such a lift will be identity. This is because the adjoint matrix of $SO(3)$ is exactly equal to the rotation matrix, i.e. $Adm\_R = R$, and $\mathfrak{so}(3)$ is isomorphic to $R^3$. We plan to explore a more general lifting network design in the near future.
>
> \[1] Hamel, Tarek, and Robert Mahony. "Attitude estimation on SO \[3] based on direct inertial measurements." _Proceedings 2006 IEEE International Conference on Robotics and Automation, 2006. ICRA 2006._. IEEE, 2006.
>
> \[2] Mahony, Robert, Tarek Hamel, and J-M. Pflimlin. "Complementary filter design on the special orthogonal group SO (3)." _Proceedings of the 44th IEEE Conference on Decision and Control_. IEEE, 2005.
>
> \[3] Deng, Congyue, et al. "Vector neurons: A general framework for so (3)-equivariant networks." _Proceedings of the IEEE/CVF International Conference on Computer Vision_. 2021.
>
> \[4] Zhu, Minghan, Maani Ghaffari, and Huei Peng. "Correspondence-free point cloud registration with SO (3)-equivariant implicit shape representations." _Conference on Robot Learning_. PMLR, 2022.
>
> \[5] Manerikar, Ninad, Minh-Duc Hua, and Tarek Hamel. "Homography observer design on special linear group SL (3) with application to optical flow estimation." _2018 European Control Conference (ECC)_. IEEE, 2018.
>
> \[6] Mei, Christopher, et al. "Homography-based tracking for central catadioptric cameras." _2006 IEEE/RSJ International Conference on Intelligent Robots and Systems_. IEEE, 2006.
>
> \[7] Hamel, Tarek, et al. "Homography estimation on the special linear group based on direct point correspondence." _2011 50th IEEE Conference on Decision and Control and European Control Conference_. IEEE, 2011.
>
> \[8] van Goor, Pieter, Tarek Hamel, and Robert Mahony. "Equivariant filter (eqf)." _IEEE Transactions on Automatic Control_ (2022).

---

> ### Author Response · Authors · 2023-11-23
> **Response to Reviewer 272k (2)**
>
> **2. As a minor comment, I found the analogy made to vector neurons more confusing than enlightening. It would be helpful to include a self-contained description of vector neurons, to make this comparison more clear.**
>
> We added a paragraph to briefly introduce Vector Neurons and compare it with our method in the paper (Sec. 4). For Vector Neurons, the input is a collection of 3D points in the Euclidean space. For our method, the input is a collection of vectors in the Lie algebra space. Each feature in our network is maintained for each input element (spatial point for VN, Lie algebra for our network), unless pooled globally. Vector Neurons is equivariant to SO(3) actions on input points, while our method is equivariant to the adjoint action of arbitrary semi-simple Lie groups on input Lie algebras. Notice that the adjoint matrix of SO(3) on its Lie algebras is the same as the rotation matrix (Adm(R)=R), and so(3) is isomorphic to R^3. Therefore, our method can be viewed as a generalization of Vector Neurons. Our method can be reduced to Vector Neurons when we identify the 3D points as so(3) elements.
>
> **3. There is no end-to-end description of the architecture... Generally speaking, the presentation could be improved — to include more details, to better motivate the equivariance problem via applications, and conditioned on this setting, to better motivate this particular architecture.**
>
> As in the response to the previous question, our architecture is largely influence by Vector Neurons. We aim to generalize the Vector Neurons to semi-simple Lie algebras because of the potential applications in robotics, computer vision, and dynamic modeling, as described in our response to question 1. We added an end-to-end illustration of the proposed method as Fig. 2 in the revised paper.
> Fig. 1 is also revised to highlight the input-output relationship. Thanks for your suggestions.

---

> ### Author Response · Authors · 2023-11-23
> **Response to Reviewer 272k (3)**
>
> **4. The only baseline included in experiments is an MLP. It would make sense to include an architecture equivariant/invariant to all linear transformations, for example, or an MLP augmented with Lie algebra transformations. On its own, the MLP seems like a weak baseline.**
>
> A linear layer can be made equivariant to all linear transformations by formulating the linear layer and the group actions at matrix multiplications from both sides of the input, as is done in Vector Neurons and our Lie networks. However, there is no existing nonlinear layer that is equivariant to all linear transformations. Nonlinear layers are the key to the expressiveness of a neural network. Thus, there is no practical architecture equivariant/invariant to all linear transformations to serve as a baseline.
>
> On the other hand, we agree that running MLP with adjoint data augmentation for comparison is a good idea. We have conducted new experiments on this setup. The results are shown in the tables below and in the paper Table 1-3. MLP with adjoint augmentation during training still largely underperforms our equivariant models.
> |               |                       |                    |                   |          |              |          |                  |          |
> | :-----------: | :-------------------: | :----------------: | :---------------: | :------: | :----------: | :------: | :--------------: | :------: |
> |               |                       | **Invariant Task** |                   |          |              |          |                  |          |
> |     Model     | Training augmentation |     Num Params     | Test augmentation |          |              |          | Invariance Error |          |
> |               |                       |                    |         Id        |          |     SL(3)    |          |                  |          |
> |               |                       |                    |        AVG        |    STD   |      AVG     |    STD   |        AVG       |    STD   |
> |      MLP      |           Id          |       136,193      |       0.148       |   0.005  |     6.493    |   1.282  |       1.415      |   0.113  |
> |      MLP      |         SL(3)         |       136,193      |       0.201       |   0.010  |     1.119    |   0.018  |       0.683      |   0.006  |
> |     LN-LR     |           Id          |       66,562       |      1.30E-03     | 3.24E-05 |   1.30E-03   | 3.25E-05 |     3.60E-04     | 5.48E-05 |
> |     LN-LB     |           Id          |       132,098      |       0.557       | 1.87E-04 |     0.557    | 1.87E-04 |   **1.43E-05**   | 1.42E-06 |
> | LN-LR + LN-LB |           Id          |       263,170      |    **8.84E-04**   | 2.52E-05 | **8.84E-04** | 2.49E-05 |     4.00E-04     |   0.000  |
>
> |         |      |             |       |          |        |     |          |    |
> | :---------------: | :-------------------: | :------------------: | :---------------: | :------: | :----------: | :------: | :----------------: | :------: |
> |          |         | **Equivariant Task** |          |          |         |          |           |          |
> |       Model       | Training augmentation |      Num Params      | Test augmentation |          |         |        | Equivariance Error |          |
> |      |       |      |         Id        |          |     SL(3)    |          |         |          |
> |      |       |      |        AVG        |    STD   |      AVG     |    STD   |         AVG        |    STD   |
> |        MLP        |           Id          |        538,120       |       0.011       | 3.53E-04 |     1.318    | 7.08E-02 |        0.424       |   0.003  |
> |        MLP        |         SL(3)         |        538,120       |       0.033       | 2.86E-04 |     0.452    | 1.01E-02 |        0.389       |   0.001  |
> |      2 LN-LR      |           Id          |        197,376       |       0.213       | 4.07E-05 |     0.213    | 4.08E-05 |      9.32E-05      | 6.65E-06 |
> |      2 LN-LB      |           Id          |        328,448       |    **9.83E-10**   | 1.78E-11 | **4.55E-08** | 8.65E-11 |    **6.56E-05**    | 4.22E-07 |
> | 2 LN-LR + 2 LN-LB |           Id          |        590,592       |      7.65E-09     | 3.54E-10 |   5.41E-08   | 4.08E-10 |      7.67E-05      | 1.56E-06 |
>
> |      |      |      |     |      |     |
> | :-----------: | :---------------------: | :----: | :---: | :-----------: | :---: |
> |            | **Classification Task** |        |       |               |       |
> |            |        Num Params       |   Acc  |       | Acc (Rotated) |       |
> |            |                         |   AVG  |  STD  |      AVG      |  STD  |
> |      MLP      |         206,339         | 95.76% | 0.65% |     36.54%    | 0.99% |
> |     LN-LR     |         134,664         | 99.56% | 0.23% |     99.51%    | 0.28% |
> |     LN-LB     |         200,200         | 99.14% | 0.21% |     98.78%    | 0.49% |
> | LN-LR + LN-LB |         331,272         | **99.62%** | 0.25% |    **99.61%**    | 0.14% |

---

> ### Author Response · Authors · 2023-11-23
> **Response to Reviewer 272k (4)**
>
> **5. What is the wedge notation ^ used in equation 4? Is this defined anywhere?**
>
> We apologize for the confusion. In this work, we focus on finite-dimensional matrix Lie algebras. The wedge is an operation that generates the Lie algebra elements using the canonical basis. Because Lie algebra is a vector space, such a basis always exists. For example, the $\mathfrak{so}(3)$ elements has the form $\begin{bmatrix}0& -\\omega_z & \\omega_y \\\ \\omega_z & 0 & -\\omega_x \\\ -\\omega_y & \\omega_x & 0 \\end{bmatrix}$. One can find the canonical basis as $E_x = \begin{bmatrix} 0& 0 & 0\\\ 0&0&-1\\\ 0&1&0 \end{bmatrix}, E_y = \\begin{bmatrix} 0& 0 & 1\\\ 0&0&0\\\ -1&0&0 \\end{bmatrix}, E_z = \begin{bmatrix} 0& -1 & 0 \\\ 1&0&0\\\ 0&0&0 \end{bmatrix}$.
> With $v=\begin{bmatrix} \omega_x, \omega_y, \omega_z \end{bmatrix}$, we have $(v)^\wedge = \omega_x E_x + \omega_y E_y + \omega_z E_z = \begin{bmatrix}0& -\\omega_z & \\omega_y \\\ \\omega_z & 0 & -\\omega_x \\\ -\\omega_y & \\omega_x & 0 \\end{bmatrix} = W \in \mathfrak{so}(3)$.
> The Vee operation is the inverse of this: $(W)^\vee = v$
> For a detailed description, we kindly refer the reviewer to Equation 10.30 and Chapter 10.2.4 in [9]. In our experiments, we use the basis for $\mathfrak{sl}(3)$ from [10].
>
> We will update the paper in Section 3.1 with this example to better convey the concept.
>
>    \[9] Chirikjian, Gregory S. _Stochastic models, information theory, and Lie groups, volume 2: Analytic methods and modern applications_. Vol. 2. Springer Science & Business Media, 2011.
>
>    \[10] Winternitz, Pavel. "Subalgebras of Lie algebras. Example of sl (3, R)." _Centre de Recherches Mathématiques CRM Proceedings and Lecture Notes_. Vol. 34. 2004.
>
> **6. I don’t understand equation 10, defining the Killing form. If I understand correctly, Adx is a function from the Lie algebra to itself, sending an input M to [X, M] = XM - MX. Adx∘Ady  is then also a function from the Lie algebra to itself. What is the trace of this function? Should we imagine setting a basis for the finite-dimensional Lie algebra, constructing the matrix that represents Adx∘Ady based on this basis, and computing the trace (which I imagine would actually be invariant to the choice of basis)?**
>
> Thank you for raising this question. The reviewer’s intuition is correct, we can compute the Killing form by finding the $ad$ operation with respect to a specific matrix. In practice, one can find a matrix, $B$, similar to a metric tensor, such that $B(x,y) = x^TBy$, where $x, y$ are the vector representation of the Lie algebra with the basis $\{E_1, E_2, \cdots, E_n\}$.
>
> An example can be found in Example 6.35 in [11] and [12]. If the reviewer is interested in a more detailed derivation, we kindly refer to Chapters 10.5.1 and 10.5.2 in [9].
>
> [9] Chirikjian, Gregory S. Stochastic models, information theory, and Lie groups, volume 2: Analytic methods and modern applications. Vol. 2. Springer Science & Business Media, 2011.
>
> [11] Kirillov, Alexander A. An introduction to Lie groups and Lie algebras. Vol. 113. Cambridge University Press, 2008.
>
> [12] https://mathworld.wolfram.com/KillingForm.html
>
> **7. Is the proposed architecture universal over continuous Lie algebra equivariant functions?**
>
> This is a great and important question. We plan to study the universal approximation theory for arbitrary width and depth cases of the proposed network in the near future.
>
> We conjecture that this is true as Euclidean spaces are one example of commutative Lie groups. At this time, we do not have the mathematical proof nor a list of regularity conditions necessary for the proof. This is an exciting and enlightening future work.
>
> **8. The authors refer to the input data as “transformations”, but also as elements of the Lie algebra. Why is it helpful to conceptualize elements of the Lie algebra as transformations? Does this interpretation still make sense for Lie groups which are not matrix Lie groups?**
>
> Thanks for the careful review of our work and your curiosity. One motivation stems from the geometric shapes of these groups. Matrix Lie groups are linear-exponential spaces as the group’s Lie algebra (tangent space at the identity) provides a vector space with bases consisting of infinitesimal group transformations known as generators. These bases are in matrix form which can define orthonormal coordinates. For example, in SO(3), these generators define three perpendicular planes where 3D rotations can be explained by simultaneous movement about the normals of these planes (i.e., a 3D coordinates frame). As such, it is a natural and perhaps less abstract concept to consider them as geometric transformations with the hope of assisting with identifying potential applications more easily.

---

> ### Author Response · Authors · 2023-11-23
> **Response to Reviewer 272k (5)**
>
> **9. How many parameters are in the baseline MLP, compared to the LN-LR and LN-LB architectures in the experiments?**
>
> We have attached the number of parameters used in each model in the table below. Our network uses a similar number of parameters as the MLP baseline.
>
> |                     |         |         |                   |
> | :-----------------: | :-----: | :-----: | :---------------: |
> |    Invariant Task   |         |         |                   |
> |         MLP         |  LN-LR  |  LN-LB  |   LN-LR + LN-LB   |
> |        136193       |  66562  |  132098 |       263170      |
> |   Equivariant Task  |         |         |                   |
> |         MLP         | 2 LN-LR | 2 LN-LB | 2 LN-LR + 2 LN-LB |
> |        538120       |  197376 |  328448 |       590592      |
> | Classification Task |         |         |                   |
> |         MLP         |  LN-LR  |  LN-LB  |   LN-LR + LN-LB   |
> |        206339       |  134664 |  200200 |       331272      |
>
> **10. As far as I understand, the adjoint operation is a particular subset of general linear transformations of the inputs. However, the linear layer is equivariant to any linear transformation, not just the adjoint operation, correct? So to clarify, are the Killing form and the Lie bracket operations special nonlinearities, in that they are only equivariant to the adjoint operation (but not to more general linear transformations)? It looks to me like the Lie bracket nonlinearity in equation (16) would be invariant general unitary transformations, but I do not understand the wedge notation, so this may not be the case.**
>
> Yes, the linear layer is equivariant to any linear transformation, while the Killing form and the Lie bracket operation are invariant and equivariant, respectively, to adjoint transformations only. Based on the definition of the wedge operation in question 5, we have the adjoint operation in two equivalent forms $(Adm_H v)^\\wedge = H(v)^\\wedge H^{-1}$. Here, $H$ is any arbitrary matrix in the Lie group, while $Adm_H$ is matrix representation of the adjoint on general linear group. The adjoint-equivariance property of the Lie bracket can then be written as $([(Adm_H x)^\\wedge, (Adm_H y)^\\wedge])^\vee=(Adm_H [x^\\wedge ,y^\\wedge])^\\vee$, or equivalently, $[Hx^\\wedge H^{-1}, Hy^\\wedge H^{-1}]=H [x^\\wedge, y^\\wedge]H^{-1}$.
>
> **11.Why is the adjoint operation meaningful to study? Does it arise often in applications? And can it not be expressed as a subgroup of GL(d), where d is the dimension of the Lie algebra?**
>
> The motivation is defining the group representation in some vector space of interest. In this work, our design choice is to work in the Lie algebra as the vector space. Then, the group representation in the Lie algebra is the adjoin representation. In this sense, it is not a choice but a natural consequence of the Lie algebraic representation. It is true that the adjoint itself forms a matrix group and, therefore, can be defined as a subgroup of the general linear group known as the matrix representation of the adjoint map (as opposed to conjugation action). In geometric problems such as control theory and robotics or computer vision/graphics where Lie groups play an important role and velocities are defined in Lie algebras, adjoint action does appear frequently. For some examples, please see [13,14,15,16,17].
>
> \[13] Barrau, Axel, and Silvère Bonnabel. "The invariant extended Kalman filter as a stable observer." IEEE Transactions on Automatic Control 62, no. 4 (2016): 1797-1812.
>
> \[14] Murray, Richard M., Zexiang Li, and S. Shankar Sastry. A mathematical introduction to robotic manipulation. CRC press, 1994
>
> \[15] Lynch, Kevin M., and Frank C. Park. Modern robotics. Cambridge University Press, 2017.
>
> \[16] Yang, Xiaolong, Xiaohong Jia, Dihong Gong, Dong-Ming Yan, Zhifeng Li, and Wei Liu. "Larnet: Lie algebra residual network for face recognition." In International Conference on Machine Learning, pp. 11738-11750. PMLR, 2021.
>
> \[17] Bayro-Corrochano, Eduardo. Geometric algebra applications vol. I: Computer vision, graphics and neurocomputing. Springer, 2018.

---

### Official Review · Reviewer_fTEN · 2023-11-01

**Soundness:** 2 fair
**Presentation:** 2 fair
**Contribution:** 2 fair
**Rating:** 6
**Confidence:** 4

**Summary:**

The paper proposes an equivariant architecture that operates on Lie algebras corresponding to transformations between vector spaces. The main contribution is the introduction of two kinds of nonlinearities: one using the Killing form of the Lie algebra and the other using the commutator. The approach is tested on two types of synthetic benchmarks as well as a classification task involving 3D geometries derived from 2D projections. The model is shown to outperform non-equivariant baselines on these tasks.

**Strengths:**

- The idea of treating elements of Lie algebras (beyond $\mathfrak{so}(3)$) as input seems novel and original, to my knowledge.
- The approach taken to construct the nonlinearities is straightforward and sensible.
- Empirical experiments appear to confirm the usefulness of such architectures.

**Weaknesses:**

- The motivation for the work is somewhat unclear. While there is one concrete example involving camera views, I would like to see additional examples to appreciate the broad applicability of this approach, even if only as citations in the introduction.

- In many useful applications, the mapping between vector spaces is not defined on the canonical basis but through representations of the group. Finding bases for these representations is a challenging task, and I think a discussion on this topic is missing.

- The fact that the Killing form for non-compact groups is not an inner product makes it ill-suited for measuring distances in the nonlinear activation. I am concerned about the potential numerical stability issues this could cause.

- The only baselines presented in the paper are standard MLPs. Although I understand that this is primarily a theoretical work, I would expect to see at least one strong baseline in one of the tasks.

**Questions:**

- I am surprised by the invariant error in Table 1 for the LN-LR model. This seems to be an order of magnitude beyond floating-point precision. Do you have any idea where this discrepancy originates?

- What is the insight behind LN-BRACKET, other than that it is an easily definable equivariant operation? It feels somewhat arbitrary, and no information is provided about the universality of a model containing this nonlinearity.

- Related to the weaknesses, can you identify a practically useful application for this approach? Such an application should have interesting baselines for comparison.

---

> ### Author Response · Authors · 2023-11-23
> **Response to Reviewer fTEN (1)**
>
> We would like to thank the reviewer for the insightful comments. Below are our response to the questions:
>
> **1. The motivation for the work is somewhat unclear. While there is one concrete example involving camera views, I would like to see additional examples to appreciate the broad applicability of this approach, even if only as citations in the introduction.**
>
>  Thanks for pointing this out. The motivation is to provide a general equivariant framework for representation learning in problems with Lie group geometric structures. The Lie algebraic representation (with adjoint group action) arises from standard group representation, making the proposed network as general as equivariance with respect to Lie group actions on vector spaces.
>
>
> We think the proposed network will have broad applications and has the potential to be adopted for applications such as 3D data processing using Inertial Measurement Units \[1,2] and Point Clouds \[3,4] where data can be viewed directly in $\mathfrak{so}(3)$ Lie algebra.
>
>
> There are applications where the inputs are transformations of semi-simple groups, notably $SL(n)$. In computer vision, $SL(3)$ is the homography group \[5,6], which contains affine(2), and 2D affine transformations as its subgroup. Representation learning on $\mathfrak{sl}(3)$ will result in homography-invariant features applicable in numerous detection and classification tasks. For physical systems, $SL(3)$ is the volume-preserving transformation (a generalization of rigid body motion $SE(3)$ in kinematics and dynamics). We conjecture the proposed work can help predict the motion of deformable shapes, even though we do not have experimental results to back up this conjecture, and this is purely our geometric intuition from analytical mechanics.
>
>  We envision our network to be applied in different applications in two ways:
>
>   1. For inputs that are directly transformations of semi-simple groups, we can directly apply our framework. This type of problems often arises in robotic state estimation and tracking, where the states being estimated are represented using elements of Lie groups. An example of this is homography tracking \[7], where measured homography matrices are noisy and fusion across multiple measurements is required.
>   2. If the measurements are raw sensor inputs which lives in some vector space, we need to find an adjoint-equivariant lift similar to Equation 13 in \[8]. This equivariant lift can be found heuristically, or it can be another equivariant neural network. For the special case where the input vectors are subjected to $SO(3)$ actions, such a lift will be the identity map. This is because the adjoint matrix of $SO(3)$ is exactly equal to the rotation matrix, i.e. $Adm\_R = R$, and $\mathfrak{so}(3)$ is isomorphic to $R^3$. To be clear, a lift from the input to the symmetry group is always possiple, e.g., via sampling, but the challenge is in constructing an equivariant lift. We plan to explore a more general lifting network design in the near future.
>
> \[1] Hamel, Tarek, and Robert Mahony. "Attitude estimation on SO \[3] based on direct inertial measurements." _Proceedings 2006 IEEE International Conference on Robotics and Automation, 2006. ICRA 2006._. IEEE, 2006.
>
> \[2] Mahony, Robert, Tarek Hamel, and J-M. Pflimlin. "Complementary filter design on the special orthogonal group SO (3)." _Proceedings of the 44th IEEE Conference on Decision and Control_. IEEE, 2005.
>
> \[3] Deng, Congyue, et al. "Vector neurons: A general framework for so (3)-equivariant networks." _Proceedings of the IEEE/CVF International Conference on Computer Vision_. 2021.
>
> \[4] Zhu, Minghan, Maani Ghaffari, and Huei Peng. "Correspondence-free point cloud registration with SO (3)-equivariant implicit shape representations." _Conference on Robot Learning_. PMLR, 2022.
>
> \[5] Manerikar, Ninad, Minh-Duc Hua, and Tarek Hamel. "Homography observer design on special linear group SL (3) with application to optical flow estimation." _2018 European Control Conference (ECC)_. IEEE, 2018.
>
> \[6] Mei, Christopher, et al. "Homography-based tracking for central catadioptric cameras." _2006 IEEE/RSJ International Conference on Intelligent Robots and Systems_. IEEE, 2006.
>
> \[7] Hamel, Tarek, et al. "Homography estimation on the special linear group based on direct point correspondence." _2011 50th IEEE Conference on Decision and Control and European Control Conference_. IEEE, 2011.
>
> \[8] van Goor, Pieter, Tarek Hamel, and Robert Mahony. "Equivariant filter (eqf)." _IEEE Transactions on Automatic Control_ (2022).

---

> ### Author Response · Authors · 2023-11-23
> **Response to Reviewer fTEN (2)**
>
> **2. In many useful applications, the mapping between vector spaces is not defined on the canonical basis but through representations of the group. Finding bases for these representations is a challenging task, and I think a discussion on this topic is missing.**
>
> Thanks for the insight. In this work, we use the standard group representation in its Lie algebra, which gives rise to the adjoint (conjugation) representation. We agree this is one particular representation of the group and if input data comes in the form of other group representation that is unknown, the current framework might not be directly applicable. At the time of developing the Lie algebraic network, we did not consider these cases and assumed the input is lifted to the group standard representation in the Lie algebra. We will add this discussion in Sec. 6 to clarify this for readers. Learning the group irreps might be a promising direction to address this limitation. https://proceedings.mlr.press/v32/cohen14.html
>
> **3. The fact that the Killing form for non-compact groups is not an inner product makes it ill-suited for measuring distances in the nonlinear activation. I am concerned about the potential numerical stability issues this could cause.**
>
> Thanks for pointing out this concern. We conducted extensive experiments on the three tasks described in the initial submission and did not observe a single case of training failure (NaN in weights).
> However, in new experiments that we added during the rebuttal (the point cloud registration task where we deal with $\mathfrak{so}(3)$-equivariance, where the network and loss functions are more complicated. More detail in Q4), we did observe training failure (NaN weights) caused by the numerical instability of the Killing-form layer.
> Our understanding of the potential risk of the Killing form not being an inner product for non-compact groups is that we cannot have the normalization term $||d||$ in the denominator, causing the output scale and thus the gradient scale to be unstable. Accordingly, we modified the Killing-form layer in the following form:
> $$
> f(x) = x, if B(x,d) <= 0;
> $$
> $$
> f(x) = x + tanh(B(x,d))d, otherwise.
> $$
> The added tanh() function constrains the Killing-form output to (-1, 1). This modified layer fixed the numerical stability problem, while the performance is consistent with models using the original Killing-form layer. We never observed training failures again. We appreciate your comment, which helped improve our network design.

---

> ### Author Response · Authors · 2023-11-23
> **Response to Reviewer fTEN (3)**
>
> **4. The only baselines presented in the paper are standard MLPs. Although I understand that this is primarily a theoretical work, I would expect to see at least one strong baseline in one of the tasks.**
>
> Unfortunately, there is no existing network equivariant to adjoint operations on semi-simple Lie Algebras. Therefore, it is not straightforward to find a comparable strong baseline. We try to address your concerns in two ways.
>
> First, we added MLP with augmentation as a stronger baseline, in which the training data undergoes $SL(3)$ adjoint action augmentation. In this way, the MLP is forced to generalize to $SL(3)$ adjoint actions, and we compare this with equivariance-by-design in our method. The results on the invariant and equivariant regression tasks are shown in the tables below. We can see that MLP with augmentations still underperforms our equivariant solutions.
>
> ****
>
> |                   |                       |                    |                   |          |              |          |                      |          |
> | :---------------: | :-------------------: | :----------------: | :---------------: | :------: | :----------: | :------: | :------------------: | :------: |
> |                   |                       | **Invariant Task** |                   |          |              |          |                      |          |
> |       Model       | Training augmentation |   Num Params   | Test augmentation |          |              |          | Invariance Error |          |
> |                   |                       |                    |       Id      |          |   SL(3)  |          |                      |          |
> |                   |                       |                    |      AVG      |  STD |    AVG   |  STD |        AVG     |  STD |
> |      MLP     |         Id        |       136,193      |       0.148       |   0.005  |     6.493    |   1.282  |         1.415        |   0.113  |
> |      MLP    |       SL(3)       |       136,193      |       0.201       |   0.010  |     1.119    |   0.018  |         0.683        |   0.006  |
> |     LN-LR     |         Id       |       66,562       |      1.30E-03     | 3.24E-05 |   1.30E-03   | 3.25E-05 |     3.60E-04     | 5.48E-05 |
> |     LN-LB     |         Id        |       132,098      |       0.557       | 1.87E-04 |     0.557    | 1.87E-04 |         **1.43E-05**        | 1.42E-06 |
> | LN-LR + LN-LB|         Id        |       263,170      |    **8.84E-04**   | 2.52E-05 | **8.84E-04** | 2.49E-05 |       4.00E-04       |   0.000  |
>
> ****
>
> |                       |                       |                      |                   |          |           |          |                        |          |
> | :-------------------: | :-------------------: | :------------------: | :---------------: | :------: | :-------: | :------: | :--------------------: | :------: |
> |                       |                       | **Equivariant Task** |                   |          |           |          |                        |          |
> |         Model         | Training augmentation |    Num Params    | Test augmentation |          |           |          | Equivariance Error |          |
> |                       |                       |                      |       Id      |          | SL(3) |          |                        |          |
> |                       |                       |                      |     AVG    |STD | AVG| STD |        AVG       | STD|
> |        MLP        |         Id        |        538,120       |       0.011       | 3.53E-04 |   1.318   | 7.08E-02 |          0.424         |   0.003  |
> |        MLP        |       SL(3)       |        538,120       |       0.033       | 2.86E-04 |   0.452   | 1.01E-02 |          0.389         |   0.001  |
> |      2 LN-LR      |         Id        |        197,376       |       0.213       | 4.07E-05 |   0.213   | 4.08E-05 |        9.32E-05        | 6.65E-06 |
> |      2 LN-LB      |         Id        |        328,448       |      **9.83E-10**     | 1.78E-11 |  **4.55E-08** | 8.65E-11 |        **6.56E-05**        | 4.22E-07 |
> | 2 LN-LR + 2 LN-LB |         Id        |        590,592       |      7.65E-09     | 3.54E-10 |  5.41E-08 | 4.08E-10 |        7.67E-05        | 1.56E-06 |
> ****

---

> ### Author Response · Authors · 2023-11-23
> **Response to Reviewer fTEN (4)**
>
> **4. The only baselines presented in the paper are standard MLPs. Although I understand that this is primarily a theoretical work, I would expect to see at least one strong baseline in one of the tasks. (Cont'd)**
>
> Second, we compare with Vector Neurons in the $\mathfrak{so}(3)$ regime. Our method can be viewed as a generalization of Vector Neurons to semi-simple Lie algebras, and mostly falls back to Vector Neurons for $\mathfrak{so}(3)$, except for the new Lie bracket nonlinear layer. We compare the performance of our network with the Lie bracket layer against the original Vector Neurons, in the point cloud classification \[3] and registration \[4] tasks. As expected, our performance is very similar to the baseline. This demonstrates that the proposed network generalizes Vector Neurons, while maintaining the performance on $SO(3)$ tasks.
>
> ****
>
> |                |                            |                          |               |
> | -------------- | -------------------------: | -----------------------: | ------------: |
> |                | Point Cloud Classification (acc, \% ) |  |Point Cloud Registration      (angular error, deg)          |
> |                |                                          |           Gaussian Noise $\sigma=0.01$  | Uniform Noise $scale=0.05$|
> | Vector Neurons |                     73.70% |                   2.2263 |        1.6305 |
> | Ours           |                     73.50% |                   2.5343 |        1.1324 |
>
> ****
>
> \[3] Deng, Congyue, et al. "Vector neurons: A general framework for so (3)-equivariant networks." _Proceedings of the IEEE/CVF International Conference on Computer Vision_. 2021.
>
> \[4] Zhu, Minghan, Maani Ghaffari, and Huei Peng. "Correspondence-free point cloud registration with SO (3)-equivariant implicit shape representations." _Conference on Robot Learning_. PMLR, 2022.

---

> ### Author Response · Authors · 2023-11-23
> **Response to Reviewer fTEN (5)**
>
> **5. I am surprised by the invariant error in Table 1 for the LN-LR model. This seems to be an order of magnitude beyond floating-point precision. Do you have any idea where this discrepancy originates?**
>
> We’re happy to explain the discrepancy. Our network indeed operates on float32 in pytorch. However, during test time, we compute the average of the error over 10,000 testing points using Python’s default precision, which is float64. Because we have $10^5$ data points, if the cumulative error across data points is small, the computed average error can be lower than float32 precision.
>
> **6. What is the insight behind LN-BRACKET, other than that it is an easily definable equivariant operation? It feels somewhat arbitrary, and no information is provided about the universality of a model containing this nonlinearity.**
>
> This is a great question that, unfortunately, we omitted further motivation in the initial draft. We have two perspectives on this definition (besides its mathematical feasibility and natural existence in the current framework):
> The residual layer with bracket has a simple explanation: it learns a delta (differential) term for the input via Lie derivative, which is the Lie bracket in the Lie algebra. In this sense, it increases the input order analogous to Taylor expansion via a first-order derivative. Higher order and nested bracket layers are also possible that we did not investigate in this work. We plan to study the universal approximation theory for arbitrary width and depth cases of the proposed network in the near future.
> The second motivation comes from potential applications of the proposed network on physical and mechanical systems. The Euler-Poincaré equations of motion in the Lie algebra are derived as [9]
> 		$$ I \dot{\xi} = [I\xi,\xi]^\vee,$$
> where $I$ is the generalized inertial matrix and $\xi \in \mathfrak{g}$ is a Lie algebra element. These are equations of motion in the Lie algebra and are equivariant. The particular $SO(3)$ case, is the classical Euler equation in the body frame for a rotating rigid body. In the future, we would like to study the application of the developed framework in learning dynamics and control. $SL(3)$ can model volume-preserving deformable bodies, a generalization of rigid body $SO(3)$, which is an exciting problem to investigate.
>
> [9] Bloch, Anthony, P. S. Krishnaprasad, Jerrold E. Marsden, and Tudor S. Ratiu. "The Euler-Poincaré equations and double bracket dissipation." Communications in mathematical physics 175, no. 1 (1996): 1-42.
>
> **7. Related to the weaknesses, can you identify a practically useful application for this approach? Such an application should have interesting baselines for comparison.**
>
> We kindly refer the reviewer to our reponses in question 1 and 4.

---

> ### Comment · Reviewer_fTEN · 2023-11-23
>
> I thank the authors for their rebuttal and appreciate the inclusion of stronger baselines, as well as a deeper insight into the LN-Bracket. While I am grateful for a better perspective on the applications provided by the authors, I still believe the evaluation, particularly in sections 5.1 and 5.2, lacks sufficient scientific value. Nevertheless, I consider this paper to be an interesting contribution to the field of equivariant neural networks for computer vision. A more robust evaluation, using a real-world dataset, could have led to a strong acceptance. In light of all this, I will maintain a weak accept and remain open to discussion with the Area Chair and other reviewers regarding the final decision.

---

### Meta-Review · Area_Chair_n8B5 · 2023-12-10

**Metareview:**

**Summary** This paper proposes Lie Neurons an architecture which operates on lie algebra inputs and is equivariant to the adjoint representation action of the corresponding Lie group.  The paper introduces two new non-linearities taking advantage of the Lie algebra structure based on the Killing form and commutator.  The method is evaluated on two synthetic benchmarks, a task classifying 3D geometry from 2D projections, and a point cloud classification task.

**Meta-Review**  The proposed architecture is novel, general for any semisimple Lie algebra, and mathematically well-motivated. The formulation treating data as elements of a Lie algebra and imposing equivariance to the adjoint action is interesting and the use of the Killing form and Lie bracket for non-linearities is clever and new.  There was broad agreement on two weaknesses with the current paper. Firstly, the paper does not make a strong case for the motivation for the method. Namely, there is no real world application demonstrated where the method makes an important impact.  Secondly, the baselines are limited to only an MLP and adjoint augmentation.  The authors argue the options for a baseline are limited due the fact other adjoint equivariant networks do not exist.  If the method were evaluated with respect to a variety of metrics in a practical application, however, then it could compete against a variety of non-equivariant baselines to demonstrate the advantage of the built in equivariance.

**Justification For Why Not Higher Score:**

- lack of motivating real world experiment
- better baselines

**Justification For Why Not Lower Score:**

N/A

---

### Decision · Program_Chairs · 2024-01-16

Reject